# INVESTESG: A MULTI-AGENT REINFORCEMENT LEARNING BENCHMARK FOR STUDYING CLIMATE INVESTMENT AS A SOCIAL DILEMMA

**Xiaoxuan Hou**[*1]**, Jiayi Yuan**[*2]**, Joel Z. Leibo**[3]**, Natasha Jaques**[2,3]
[1]Foster School of Business, University of Washington
[2]Paul G. Allen School of Computer Science and Engineering, University of Washington
[3]Google Deepmind
xxhou@uw.edu, jiayiy9@cs.washington.edu, jzl@google.com
nj@cs.washington.edu

## ABSTRACT

**InvestESG** is a novel multi-agent reinforcement learning (MARL) benchmark designed to study the impact of Environmental, Social, and Governance (ESG) disclosure mandates on corporate climate investments. The benchmark models an intertemporal social dilemma where companies balance short-term profit losses from climate mitigation efforts and long-term benefits from reducing climate risk, while ESG-conscious investors attempt to influence corporate behavior through their investment decisions. Companies allocate capital across mitigation, greenwashing, and resilience, with varying strategies influencing climate outcomes and investor preferences. We are releasing open-source versions of InvestESG in both PyTorch and JAX[1], which enable scalable and hardware-accelerated simulations for investigating competing incentives in mitigate climate change. Our experiments show that without ESG-conscious investors with sufficient capital, corporate mitigation efforts remain limited under the disclosure mandate. However, when a critical mass of investors prioritizes ESG, corporate cooperation increases, which in turn reduces climate risks and enhances long-term financial stability. Additionally, providing more information about global climate risks encourages companies to invest more in mitigation, even without investor involvement. Our findings align with empirical research using real-world data, highlighting MARL's potential to inform policy by providing insights into large-scale socio-economic challenges through efficient testing of alternative policy and market designs.

## 1 INTRODUCTION

Climate change poses a persistent threat to global stability, with droughts, storms, fires, and flooding becoming more intense and frequent (Christopher B Field & Dahe, 2012), leading to disruption of the natural ecosystem and significant impacts on the global economy. Addressing climate change requires coordinated efforts across multiple sectors, particularly from large corporations, which are reportedly responsible for over 70% of global industrial greenhouse gas emissions (Griffin & Heede, 2017). While adaptation—preparing for the inevitable consequences of climate change—tends to be party-specific and often driven by financial incentives, mitigation—reducing emissions—presents a an intertemporal social dilemma (Leibo et al., 2017; Hughes et al., 2018), where the benefits of reduced emissions are shared globally yet the costs are borne locally (Olson Jr, 1971; Dahlman, 1979; Buchanan & Stubblebine, 2006). As corporations are inherently self-interested, they are unlikely to reduce emissions voluntarily without external incentives or regulations.

Numerous policies have been proposed to address this challenge. Among these, mandatory Environmental, Social, and Governance (ESG) disclosures have recently been hotly debated in the United States. The Securities and Exchange Commission's (SEC) ESG proposal, which would require publicly traded companies to disclose climate-related risks, mitigation strategies, and greenhouse gas

---

[1]Github repo: https://github.com/yuanjiayiy/InvestESG

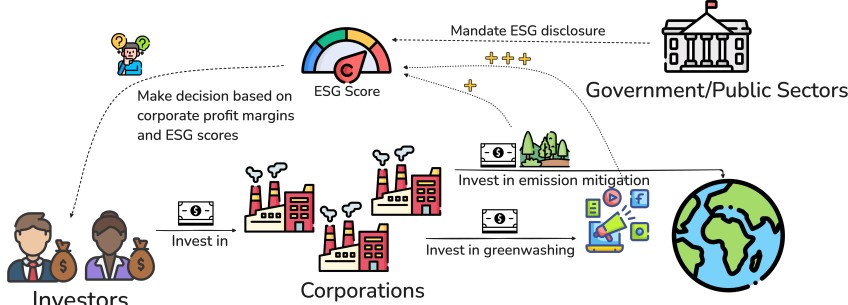

Figure 1: The **InvestESG** Environment. Corporations choose how much to invest in mitigating emissions, which affects their ESG Score. Climate-conscious investors can see ESG Scores when deciding how much to invest in each company. However, companies can engage in greenwashing to inexpensively and falsely improve ESG scores without actually mitigating climate change. InvestESG is a social dilemma, where selfish, profit-motivated corporations will not invest in mitigation without further incentives, leading to increased climate risks and decreased global wealth.

emissions from their operations, has attracted over 15,000 comments, making it one of the most contentious proposals in the SEC's history (SEC, 2024b; CNBC, 2024). This has resulted in an indefinite delay in enactment of the policy to allow for further discussion (SEC, 2024a). The U.S. is not alone in facing such pushback; similar delays are unfolding in the European Union and Korea (Bloomberg News, 2024; Korea Economic Daily, 2023). This highlights the need for thorough research to effectively inform the design and implementation of these policies.

Traditional economics and policy research relies on either empirical analysis—which does not enable testing possible new policies (Doshi et al., 2013; Li & Wu, 2020; Krueger et al., 2021)—or theoretical economics models, which are often limited to scenarios with only two agents (e.g., Friedman et al. 2021), or single-period games (e.g., Pástor et al. 2021). In contrast, Multi-Agent Reinforcement Learning (MARL) enables simulating complex interactions between multiple agents over extended time periods, under diverse hypothesized policy settings. Leveraging MARL to address large-scale socio-economic questions is a growing field (Hertz et al., 2023). Prior work has demonstrated the potential of MARL to design effective taxation schemes that enhance both equality and productivity (Zheng et al., 2021), highlighting its relevance for tackling social challenges.

We propose using multi-agent reinforcement learning (MARL) to explore the impact of the ESG disclosure policy. We introduce **InvestESG**, an open-source MARL benchmark, to examine how profit-driven corporations balance short-term profits with long-term climate investments and whether ESG-informed investor choices influence corporate behavior. The simulation involves two agent types: companies and investors. Companies allocate funds to mitigation, greenwashing, and resilience, while investors choose investment portfolios based on their preferences for financial returns versus ESG benefits. This creates an intertemporal social dilemma (Hughes et al., 2018), where both agent classes must weigh immediate and local costs against long-term and global gains. Using Schelling diagrams, we demonstrate that in a fully profit-driven environment, a social dilemma arises, but with sufficiently ESG-conscious investors with enough capital, climate mitigation becomes optimal for some or all corporations. However, if companies are able to *greenwash* to cheaply increase ESG scores without genuine mitigation, the environment once again becomes a social dilemma.

Our experiments with a state-of-the-art MARL implementation yield findings that align with real-world empirical evidence and provide novel insights. First, a sufficient number of highly ESG-conscious investors are needed to incentivize corporate mitigation efforts. In such cases, climate-focused companies emerge and attract most climate-conscious investments, while others prioritize profit maximization. When only some investors are climate-focused, the market bifurcates, with mitigating companies aligning with conscious investors. Additionally, sharing climate risk information helps companies to increase mitigation efforts, even when investors are not present.

More broadly, we demonstrate the potential of using a MARL framework to inform policy debates in the field of climate change. Importantly, our results are compatible broadly with the existing theoretical and empirical literature that examines this question, which shows its validity in predicting directional behaviors of rational decision makers. Assessing the effectiveness of a policy is

inherently challenging, due to the fact that policy experiments are often prohibitively expensive and impractical to conduct, and even when they are feasible, it can be extremely time-consuming. Given the urgency of addressing climate change, our work provides a new vector for studying this problem, creating a simulated environment where a broad range of regulations can be explored and tested efficiently to provide novel insights into the problem.

Our aim is not to claim that our model fully represents real-world complexity. It remains a "first-principles" model; however, it captures aspects that are challenging to integrate into a single framework using a traditional economic approach. We present InvestESG as a challenge for the machine learning community; for researchers interested in developing better cooperative agents, or improved MARL algorithms, InvestESG represents a benchmark that could actually inform policymaking.

## 2 RELATED WORK

Creating a benchmark environment that analyzes the interplay between corporations and investors and their impacts on climate change mitigation requires various domain knowledge and connects multiple streams of studies. We closely examined three streams of literature.

**Conventional economic methods are limited by either generalizability or tractability.** Existing ESG disclosure research in economics, business, and public policy rely on either empirical data (Doshi et al., 2013; Li & Wu, 2020; Krueger et al., 2021) or simplified theoretical models (Polinsky & Shavell, 2012; Kalkanci & Plambeck, 2020; Cho et al., 2019; Pástor et al., 2021; Friedman et al., 2021). Empirical analyses, while grounded in real-world data, struggle with generalizability and testing counterfactual policies. Theoretical models provide formalized equilibria and explore counterfactuals but are limited by tractability, modeling either multiple agents in single-period games (e.g., Pástor et al. 2021) or only two agents over limited time periods (e.g., Friedman et al. 2021). By proposing a MARL framework, our method overcomes these limitations by enabling the simulation of complex, multi-agent systems over extended time horizons under diverse policy settings, which allows for emergent behaviors and better captures complex socio-economic dynamics among diverse agents (Hertz et al., 2023).

**Current MARL benchmarks and social dilemma environments were not designed to model specific policy problems.** Various MARL benchmarks have been created to study multi-agent coordination and cooperation; however, they are often limited to simplistic particle simulations (Lowe et al., 2017) or videogames (Agapiou et al., 2023; Carroll et al., 2020; Samvelyan et al., 2019; Bettini et al., 2024b;a), which have little direct real-world implication. Sequential social dilemmas (SSD) (Leibo et al., 2017) are spatially and temporally extended multi-agent environments in which the payoff to an individual agent for defecting is higher, but if all agents defect the payoff is lower. SSDs go beyond traditional game-theoretic environments like Prisoner's Dilemma because the complexity of the solving the SSD depends on not only addressing the misalignment between individual and collective rationality, but doing so when the negative consequences of short-sighted actions may take a long time to manifest. Prior research has examined methods to promote cooperation in SSDs by incorporating inequity aversion in agents (Hughes et al., 2018), rewarding agents for influencing others' actions (Jaques et al., 2018), enabling agents to provide incentives to others (Yang et al., 2020a), and controlling for behavioral diversity (Bettini et al., 2024a). Many of these studies are inspired by factors that drive human cooperation in social dilemmas. However, much of this research has been conducted in environments with no direct real-world implications (e.g. Leibo et al. 2017; 2021). In contrast, our environment directly addresses the problem of climate change, and is designed with critical trade-offs around a specific policy question. We hope to encourage researchers interested in addressing social dilemmas to focus on an environment with potential beneficial social impact on the problem of climate change (Bisaro & Hinkel, 2016).

**RL benchmarks can be effective in focusing AI research on climate change issues.** Learning to Run a Power Network (L2RPN) is a single-agent RL benchmark focused on improving power grid efficiency (Marot et al., 2021). This work has spawned multiple competitions, and it is currently hosted by the Electric Power Research Institute (EPRI) in conjunction with several other energy companies, government agencies, and universities[2]. L2RPN continues to foster innovative and meaningful collaboration across institutions, showing the potential impact of this type of sim-

---

[2]https://www.epri.com/l2rpn

plified, simulated RL benchmark. However, L2RPN is focused on single-agent RL, whereas we explore a multi-agent, multi-party social dilemma. The only other MARL climate benchmark we are aware of was proposed by Zhang et al. (2022), and is focused on studying the dynamics of international climate negotiations, by incorporating an integrated assessment model simulating global climate and economic systems. In contrast, our environment, InvestESG, was designed with a focus on a more targeted policy question regarding ESG disclosure mandates, making our approach more directly applicable to an ongoing and highly debated policy issue.

## 3 PROBLEM SETTING

Corporations face two key trade-offs in climate action. The first is between short-term costs and long-term benefits, as climate efforts often require upfront investments with returns realized over time. The second is between self-interest and collective benefit, as emission reductions benefit all, but the costs are borne by those who act, which incentivizes free-riding and discourages collective action. These trade-offs motivate us to conceptualize corporate climate actions as an intertemporal social dilemma (Leibo et al., 2017; Bisaro & Hinkel, 2016), which by definition, involves a conflict between immediate self-interest and longer-term collective interests.

We introduce InvestESG, a MARL benchmark environment designed to evaluate the impact of ESG disclosure mandates on corporate climate investments. The environment is grounded in well-established finance and economics literature that explores similar questions, providing a conceptual framework for studying corporate and investor dynamics in response to ESG policies. Pastor et al. (2021) and Pedersen et al. (2021) proposed analytical models of single-period equilibria, where companies either choose or are assigned fixed levels of "greenness," and investors optimize portfolios to balance financial returns with preferences for green assets. We follow a similar setting where companies choose mitigation effort levels every period and receive a single metric ESG score that reflects their greenness. Investors derive utility from the profits generated by their portfolios and, based on their level of "ESG-consciousness", may also receive positive utility by investing in firms with high ESG scores. We extend these frameworks by (1) modeling the evolution of firm-investor strategies over the long run, which poses significant analytical and numerical challenges (Pakes & McGuire, 2001), (2) incorporating climate evolution, where risks worsen over time but can be mitigated by firm efforts, and (3) introducing greenwashing (Lyon & Maxwell, 2011) where companies cheaply inflate ESG scores without genuine mitigation, which raises issues of information asymmetry and principal-agent problems (Grossman & Hart, 1992).

We aim to address the following research questions: (1) Do ESG-conscious investors incentivize companies to undertake mitigation efforts, and how does the level of ESG-consciousness influence these outcomes? (2) Does the presence of heterogeneous investors lead to bifurcation in agent strategies, where only some companies engage in mitigation to attract ESG-conscious investors, while others and their investors opt to free-ride? (3) Are companies likely to engage in greenwashing? (4) What measures can enhance the effectiveness of ESG disclosure mandates?

## 4 THE INVESTESG ENVIRONMENT

In this section, we provide a brief overview of the environment. The detailed mathematical formulation is presented in Appendix 9. The simulation starts in 2021 and runs through 2120, with each period $t$ corresponding to one year. The environment includes two main components: (1) an evolving climate and economic system, and (2) two types of agents: $M$ company agents $\mathcal{C}_i$ for $i \in \{1, \ldots, M\}$ and $N$ investor agents $\mathcal{I}_j$ for $j \in \{1, \ldots, N\}$.

**Climate and Economic Dynamics.** The environment is initialized with a climate risk parameter, $P_0$, representing the probability of at least one adverse climate event occurring. The value of $P_0$ is benchmarked against Masson-Delmotte et al. (2021), reflecting the likelihood of extreme heat, heavy precipitation, and drought events. In the absence of mitigation efforts, climate risk increases over time, aligning with the IPCC's $4\,^\circ\mathrm{C}$ warming scenario. Figure 2a depicts how increased climate risks and adverse climate events increase over time in a scenario where companies are solely profit-motivated. Company agents can mitigate the growth of climate risk by investing in emissions *mitigation*. Climate events are modeled as Bernoulli processes, with their probabilities determined by the climate risk of the corresponding year. Multiple events can occur within a single year, as illus-

trated by the red dashed lines in Figure 2a. In addition to the evolving climate risks, the environment incorporates a baseline *economic growth rate* $\gamma$, set to 10% by default, aligned with the historical average annual return of the S&P 500 over the past century (Damodaran, 2024a). Company agents' capital levels $K_t^{\mathcal{C}_i}$ grow at rate $\gamma$ each year, barring climate events. If an adverse event occurs, company agents lose a portion of their total capital according to their respective *climate resilience* parameter $L_t^{\mathcal{C}_i}$. If economic losses drive a company's remaining capital into negative territory, the company is declared bankrupt.

**Action Space.** Each company agent $\mathcal{C}_i$ in period $t$ selects actions from a continuous vector $\mathbf{u}_t^{\mathcal{C}_i} = (u_{t,m}^{\mathcal{C}_i}, u_{t,g}^{\mathcal{C}_i}, u_{t,r}^{\mathcal{C}_i})$, where $u_{t,m}^{\mathcal{C}_i}$ represents the share of capital allocated to *mitigation*, $u_{t,g}^{\mathcal{C}_i}$ to *greenwashing*, and $u_{t,r}^{\mathcal{C}_i}$ to building climate *resilience*.

Each investor agent first selects the set of companies it will invest in. Specifically, agent $\mathcal{I}_j$ in period $t$ selects an action from a binary vector of length $M$, $\mathbf{a}_t^{\mathcal{I}_j} = (a_{t,1}^{\mathcal{I}_j}, \ldots, a_{t,M}^{\mathcal{I}_j})$, corresponding to the $M$ company agents.

**Modeling ESG Disclosure.** With the ESG disclosure mandate in place, each company agent receives an updated ESG score $Q_{t+1}^{\mathcal{C}_i}$ in period $t$, calculated as $Q_{t+1}^{\mathcal{C}_i} = u_{t,m}^{\mathcal{C}_i} + \beta u_{t,g}^{\mathcal{C}_i}$, based on mitigation and greenwash spending, where $\beta > 1$ indicates that greenwashing is cheaper than genuine mitigation in terms of building an ESG-friendly image.

**State and Observation Space.** The environment simulates a partially observable Markov game $\mathcal{M}$ defined over a continuous, multi-dimensional state space. The system state at period $t$ is characterized by the climate risk parameter $P_t$, each company agent's state vector, consistent of its capital level, its ESG, and its climate resilience. Each investor agent's state is represented by their investment portfolio and cash levels. All company and investor agents share a common observation space, with the state of all companies and investors. Extensions that incorporate additional observable information, like climate risk, are explored in Section 5.

**State Transition.** At the beginning of period $t$, investors collect their investment holdings from period $t-1$ and redistribute their capital according to $\mathbf{a}_t^{\mathcal{I}_j}$. If an investor opts not to invest, all capital remains as cash. Companies then make climate-related spending, after which system climate risk and companies' individual resilience are updated. Meanwhile, companies receive updated ESG scores, and the occurrence of climate events are simulated. Afterwards, companies' profit margins, $\rho_t^{\mathcal{C}_i}$, are computed as a function of their climate-related spending $\mathbf{u}_t^{\mathcal{C}_i}$, default economic growth $\gamma$, and losses due to climate events, $L_t^{\mathcal{C}_i}$, and the number of climate events $X_t$. Finally, company capitals are updated, and so are investor holdings and cash based on the profit margins of their portfolios.

**Rewards.** The single-period reward for company $\mathcal{C}_i$ is solely based on its profit, reflecting the assumption that companies are profit-driven. The reward for investor $\mathcal{I}_j$ is the summation of two components: (1) the portfolio profit margin, and (2) the weighted average ESG score of the investor's portfolio multiplied by the investor's ESG preference, $\alpha^{\mathcal{I}_j}$.

**Social Outcome Metrics.** We evaluate agent performance based on two key social outcome metrics: the final climate risk level, $P_{100}$, and the total market wealth at the end of the period, $W_{100}$. Additionally, we keep track of other social outcome metrics such as total number of severe climate events and number of companies that go bankrupt.

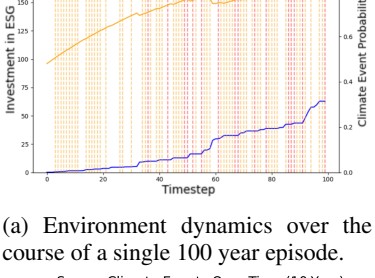

(a) Environment dynamics over the course of a single 100 year episode.

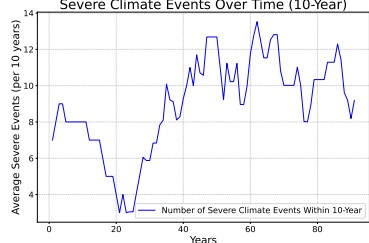

(b) Average number of severe climate event occurrences over 10-year period for the scenario in part (a).

Figure 2: Status quo scenario where all agents are only profit-motivated. In (a), mitigation spending (blue curve) is minimal, leading climate risk (yellow curve) to increase over time. Adverse weather event occurrences are shown as dotted lines; red lines indicate multiple adverse events in a single year. (b) plots the average number severe climate events over the episode in (a), showing how increasing climate risk leads to more frequent extreme weather events.

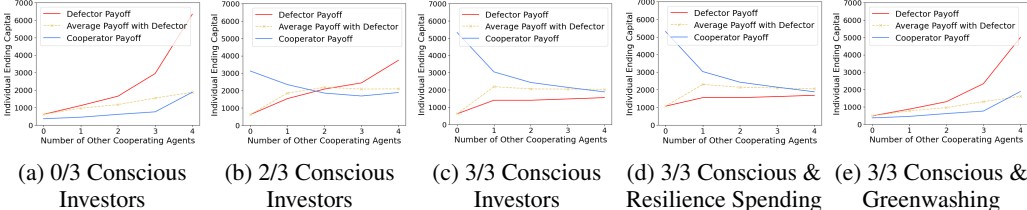

| (a) 0/3 Conscious Investors | (b) 2/3 Conscious Investors | (c) 3/3 Conscious Investors | (d) 3/3 Conscious & Resilience Spending | (e) 3/3 Conscious & Greenwashing |

Figure 3: Schelling diagrams demonstrating that the environment constitutes a social dilemma. The graphs compare payoffs between cooperation (mitigation, blue lines) and defection (no mitigation, red lines) for a focal company, given varying number of other cooperating companies. Yellow lines represent the average payoff across all companies when the focal company defects. Subfigure (a) illustrates the selfish scenario, where all three investors consistently prioritize financial returns ($\alpha^{\mathcal{I}_j} = 0$, for $j = 1, 2, 3$). Here, defection always yields higher payoffs for the focal company than cooperation, leading all companies to defect. However, widespread defection results in lower overall profits, as the average payoff (yellow) increases with greater cooperation, demonstrating the environment constitutes a social dilemma. Subfigure (b) and (c) correspond to two and three infinitely ESG-conscious investors ($\alpha^{\mathcal{I}_j} \approx \infty$), respectively. In (b), cooperation yields higher payoffs than defection for the focal company when few others cooperate. In (c), cooperation outperforms defection in all cases. Therefore, (b-c) demonstrate how investor behavior can transform the environment, eliminating the social dilemma by aligning corporate incentives with mitigation. Subfigures (d) and (e) build on (c) with three ESG-conscious investors. Subfigure (d) introduces resilience spending, while (e) adds greenwashing. The latter reintroduces a social dilemma, where corporations again avoid mitigation.

## 4.1 THE TRADE-OFF FOR THE AGENTS FRAMES CLIMATE CHANGE AS A SOCIAL DILEMMA

Companies' trade-off between short-term private costs and long-term collective benefits create a social dilemma within the environment. To illustrate this, we use empirical Schelling diagrams (Hughes et al., 2018), which plot the payoff of following either a cooperative or defecting policy, depending on the number of other cooperating company agents in a 5-company, 3-investor scenario.

In our case, cooperation represents altruistic investment in mitigating emissions. We simulate a co-operative policy as one which consistently invests 0.5%[3] of capital in mitigation and 0 in greenwashing or resilience building. In Figure 3a to 3c, a defector policy takes zero action in all of mitigation, greenwashing, and resilience. As shown in Figure 3a, when investors are solely profit-driven, the payoff for following a defector policy is always higher than the cooperator payoff. However, if all agents fail to cooperate, overall payoffs will be extremely low, showing that the environment represents a social dilemma. Figures 3b and 3c illustrate how the payoffs for cooperation and defection evolve as the proportion of *ESG-conscious investors* with positive ESG preferences increases. Notably, Figure 3b suggests the possibility of a bifurcated equilibrium when investor preferences are mixed. Specifically, climate-friendly companies may attract ESG-conscious investors, while other companies free ride and attract profit-motivated investors. Figure 3c shows that when all investors prioritize ESG-friendly firms, cooperation dominates defection in all cases. Therefore, with enough ESG-conscious investors the environment is no longer a social dilemma, and companies are actually greedily motivated to invest in mitigation to attract investors.

**Greenwashing and Climate Resilience Investment.** Although Figure 3c appears promising, the option to greenwash and spend on resilience complicates the trade-offs companies face. In Figure 3e, we define the defection policy as investing 0.5% of capital annually in resilience. This makes defection slightly more attractive compared to Figure 3c, as resilience spending results in higher capital gains than taking no action. Figure 3d presents an alternative scenario where defecting companies invest 0.3% annually in greenwashing[4]. In this case, cooperation becomes significantly more challenging, because companies can attract ESG-conscious investors without actually mitigating their emissions. The addition of greenwashing turns the environmental back into a social dilemma.

---

[3]Since company actions are continuous, we selected 0.5% as a benchmark for illustrating the social dilemma structure in the Schelling diagram. This aligns with real examples of cooperative corporations, such as Patagonia, where ∼0.3% of its valuation is allocated annually to climate initiatives. The diagram's shape remains consistent across other tested values.

[4]For the Schelling diagram, we set the greenwashing coefficient $\beta = 2$. As a result, 0.3% spending on greenwashing leads to a higher ESG score than 0.5% spent on mitigation.

Figure 4: Ending values for all metrics averaged over the last 100 episodes; error bars show std. err. over 3 random seeds. We compare the status quo scenario with solely profit-driven investors (investors with ESG consciousness level of 0), both with and without the ESG disclosure mandate, to scenarios involving three ESG-conscious investors with ESG consciousness level of $\alpha = 0.5$, $\alpha = 1$, and $\alpha = 10$. These results indicate that merely disclosing ESG scores is insufficient to resolve the social dilemma if investors are not interested in investing in climate-friendly companies. However as investors' level of ESG consciousness increases, the ESG mandate results in consistent improvements in mitigation, climate risk, and market wealth.

## 4.2 MULTI-AGENT REINFORCEMENT LEARNING BASELINES

We are motivated to simulate companies and investors as independent, selfishly motivated agents that specialize in maximizing their own expected reward. We employ the state-of-the-art Independent PPO (IPPO) algorithm (De Witt et al., 2020; Yu et al., 2022) so each agent has its own policy parameters, and agents do not share parameters among themselves. InvestESG is implemented in both PyTorch (Paszke et al., 2017) and JAX (Bradbury et al., 2018), capable of simulating the investment behaviors in a massively parallel sense by vectorization on both CPUs and GPUs. By default, 5 companies are modeled with an initial capital of $10 trillion each, alongside 3 investors, each starting with $16 trillion. This creates a total beginning market wealth of $98 trillion, which is comparable to the global stock market cap (WFE Statistics, 2023). We conduct the experiments in a 5-company, 3-investor environment by default unless specified otherwise. All the following results are averaged based on 3 runs of different random seeds, with the standard error plotted in the shaded region [5]. Following economic literature that argues more than 10 agents are often needed to observe meaningful group size effects (Arifovic et al., 2023), we provide additional results in Appendix 11 from settings with 10 companies and 10 investors, as well as 25 companies and 25 investors, which are consistent with our findings with the 5-company, 3-investor setting.[6]

## 5 MAIN RESULTS

**Level of investors' ESG-consciousness increases mitigation efforts.** To evaluate the effectiveness of ESG disclosure mandates in incentivizing emissions mitigation by companies, we conduct three experiments: (1) *Status Quo*: All agents are profit-motivated, and no ESG scores are released; (2) *Status Quo with Mandate*: ESG scores are disclosed, but investors remain profit-driven ($\alpha = 0$); (3) *Mandate with ESG-Conscious Investors*: ESG scores are disclosed, and investors are ESG-conscious ($\alpha > 0$). In this first set of experiments, we study the default setting in which both greenwashing and resilience spending are disabled.

The final system climate risks are shown in Figure 4. The values for *Status Quo with Mandate* and *Status Quo* are similar, indicating that mandatory ESG disclosure alone does not significantly incentivize mitigation when investors prioritize profits. However, we see that mitigation spending increases consistently corresponding to each increase in ESG-consciousness level ($\alpha = 0.5, 1, 10$), leading to corresponding improvements climate risks and market wealth, with the best values

---

[5]In simulating climate events as described in Equation 2, we use a fixed random seed across training episodes. This reduces stochasticity while ensuring event occurrence varies with episode-specific climate risks. This design choice is based on two considerations. First, unlike MARL agents learning from scratch, real-world companies possess a baseline understanding of trade-offs and can reasonably anticipate future risks; a more predictable training environment mirrors this insight. Second, our study aims not to develop agents that solve climate change under all stochastic scenarios, but rather to efficiently train agents that grasp the strategic trade-offs faced by companies and investors. Limiting stochasticity makes learning and convergence easier without compromising the study's structural conclusions. This design is also consistent with classic versions of climate-economic simulation models like DICE and RICE, where the link between economic development and $CO_2$ mass and the damage functions of global warming are largely deterministic (Nordhaus, 1992; Nordhaus & Yang, 1996; Nordhaus, 2017). The implementation also includes the option to not use a fixed random seed.

[6]The 25-by-25 agent setting aligns with Zhang et al. (2022), which employs 27 agents in a MARL framework to address climate change issues.

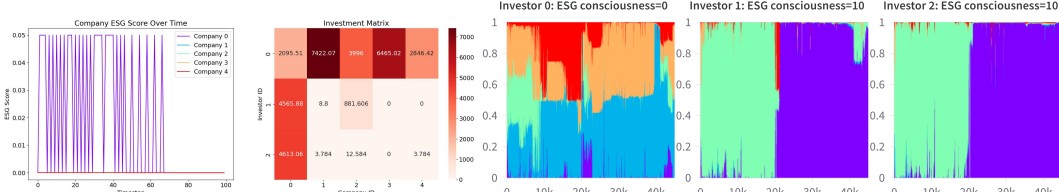

(a) Investment matrix at the latest episode for investors with different level of ESG consciousness: 0, 10, 10.

(b) Investors with different level of ESG consciousness make different investment decisions.

Figure 5: Investigating the effects of the level of ESG consciousness in the case of 5 companies and 3 investors, where investor 0 is profit driven ($\alpha^{\mathcal{I}_0} = 0$), and investors 1 and 2 are deeply climate-conscious ($\alpha^{\mathcal{I}_1} = \alpha^{\mathcal{I}_2} = 10$). In (a) and (b), Company 0 (purple) learns to be the leading mitigator. The figures plot the investment distribution for each investor, showing that more climate-conscious investors focus on investing in the more climate-conscious companies, mirroring the market bifurcation results of the Schelling diagrams.

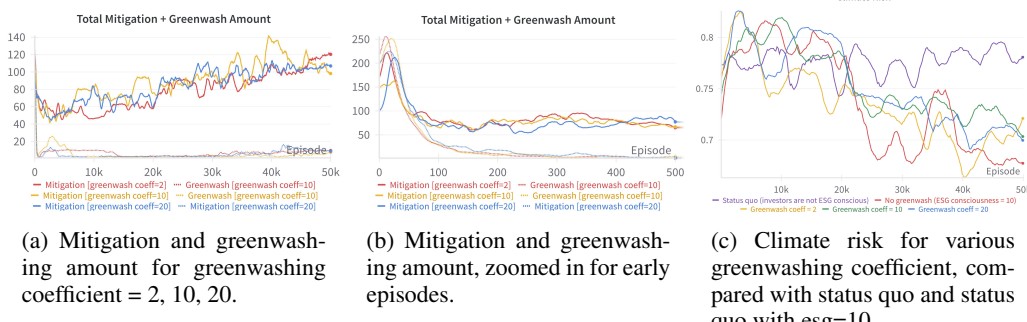

(a) Mitigation and greenwashing amount for greenwashing coefficient = 2, 10, 20.

(b) Mitigation and greenwashing amount, zoomed in for early episodes.

(c) Climate risk for various greenwashing coefficient, compared with status quo and status quo with esg=10.

Figure 6: Mitigation, greenwashing, and climate risk for greenwashing coefficient = 2, 10, 20, with initial exploration episodes zoomed in in (b). It shows that when both companies and investors are IPPO agents, regardless of greenwashing cost, companies initially explore greenwashing and mitigation equally but quickly abandon greenwashing and invest in mitigation to attract ESG-conscious investors. In these experiments, the final climate risk is similar to the value when greenwashing is not enabled, suggesting greenwashing does not hinder mitigation investment.

achieved for $\alpha = 10$. In this setting, a few leading companies attract the majority of ESG-conscious investments, forming a positive feedback loop where better market performance draws further investments (see Appendix Figure 10c and 10d for a demonstration for such pattern in the 10 companies, 10 investors case). These findings align with research showing that ESG disclosure mandates encourage climate-friendly efforts driven by societal and stakeholder pressures (Cormier & Magnan, 1999; Gamerschlag et al., 2011; Fama & French, 2007; Friedman & Heinle, 2016; Chen et al., 2018; Ioannou & Serafeim, 2019), and that fund managers are willing to sacrifice financial returns for ESG benefits (Krueger et al., 2021).

**Heterogeneous investor preferences lead to bifurcation in agent strategies.** Research shows that investors have varying preferences for ESG efforts and respond in different ways (Amel-Zadeh & Serafeim, 2018). In the scenario plotted in Figure 5, we initialized three investors with different levels of ESG-consciousness, with Investor 0 representing the solely profit-seeking investor ($\alpha^{\mathcal{I}_0} = 0$), and Investor 1 and 2 representing highly ESG-conscious investors ($\alpha^{\mathcal{I}_1} = \alpha^{\mathcal{I}_2} = 10$), which corresponds to the scenario presented in the Schelling diagram in Figure 3b. Figure 6a and 6c highlight a divergence in both investor and company behavior. Profit-driven Investor 0 distributes investments evenly across companies, while ESG-focused investors (1 and 2) favor climate-conscious firms, such as Company 0, which prioritizes mitigation. Such divergence extends to company strategies: Company 0 learns to attract more ESG investment by focusing on mitigation, while others prioritize financial returns at the expense of some investor interest.

**The possibility of greenwashing does not significantly undermine mitigation efforts.** Building on research that examines the existence and extent of greenwashing (Wu et al., 2020; Marquis et al., 2016; Siano et al., 2017), and the concern that the ESG disclosure policy can backfire with greenwashing (El-Hage, 2021), we explore whether companies adjust their strategies when greenwashing is permitted. Based on the Schelling diagram in Figure 3d and Figure 6c, we predict that investors would be misled by greenwashing, prompting companies to prioritize greenwashing over mitigation

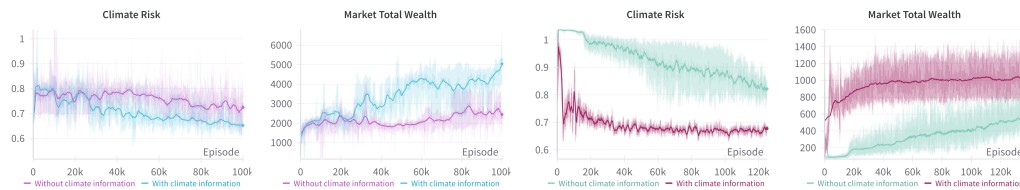

(a) More information for investors and companies          (b) More information for companies (no investors)

Figure 7: Effect of providing more information about climate risk to both investors and companies in the default 5-company-3-investor case (a), or companies only in a 5-company-0-investor case (b). These results show that simply providing more information about climate risk to companies can help them coordinate to increase mitigation efforts.

due to its lower cost, leading to waste of resources. To verify greenwashing can attract climate-conscious investors in our simulation, we hard-code companies to follow either a greenwashing or mitigation strategy. Figure 9 shows how in this scenario the greenwashing company earns a higher ESG score and succeeds in attracting climate-conscious investors. However, this trend is not observed when simulating agents with IPPO. Figure 6 plots results under varying greenwashing costs, represented by coefficient $\beta$ in Equation 3, where a larger $\beta$ indicates cheaper greenwashing. All investors have an ESG-consciousness level of $\alpha = 1$. The results show that regardless of greenwashing cost, companies initially explore greenwashing and mitigation equally but quickly drop greenwashing as the main strategy. This likely occurs because, in the early episodes, investors have not yet linked ESG scores to their investment strategies, so do not respond to greenwashing efforts with increased investment. Instead, greenwashing presents an immediate cost for companies, which does not pay off with reduced climate risk, so they quickly learn to avoid it. In contrast, even without immediate investor rewards, companies may recognize the long-term benefits of mitigation (which leads to higher collective returns), and be incentivized to continue those efforts at first, even if they later learn to defect. This mirrors reality, where if investors are slow to adjust their strategies, companies may abandon greenwashing early but persist in low levels of mitigation for either its long-term climate benefits or the foresight of regulations and societal pressure that will eventually nudge them towards sustainable operation. This is consistent with empirical literature that despite the emergence of some greenwashing, ESG disclosure mandates overall encourage mitigation (Fiechter et al., 2022).

**Providing additional information about climate risk increases mitigation efforts.** In this scenario, we test whether providing additional climate-related information to agents can affect mitigation behavior. Therefore, we provide the climate event probability and climate event occurrences as additional information in the observation space for both companies and investors. In the default 5-company-3-investor scenario, having additional climate-related information reduces the ending system climate risks, as shown in Figure 7a. Figure 7b tests a scenario with no investors, in which companies are provided with the same additional climate-related information. Here climate information leads to a large reduction in climate risk, with the final risk ($\approx 0.7$) comparable to the risk achieved in the $\alpha = 1$ scenario in Figure 4. This result suggests that climate-related information helps both companies and investors make better environmental decisions. Even in the absence of ESG-focused investors, educating companies about the overall system can significantly improve outcomes. This aligns with literature showing that raising awareness encourages positive corporate responses (Delmas & Toffel, 2008; Bowen, 2000).

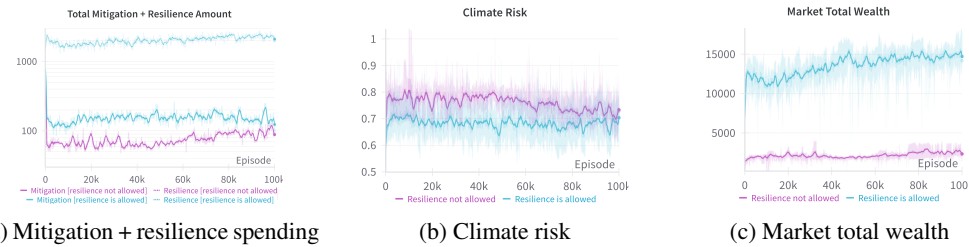

(a) Mitigation + resilience spending          (b) Climate risk          (c) Market total wealth

Figure 8: Effect of allowing resilience spending. When resilience is allowed, companies invest significantly more in resilience than in mitigation, resulting in ending episode higher market wealth, and lower climate risk.

**Effects of resilience spending.** In this case, we allow companies to invest in resilience spending to improve their own robustness to climate events, without affecting the global climate risk. Although

investors continue to respond to companies' ESG scores, resilience spending is excluded from the calculation of ESG scores. As shown in Figure 8, when resilience is allowed, companies invest significantly more in resilience than in mitigation. By investing in resilience, companies are more resilient to the climate event and therefore are able to maintain more capital to invest in actual mitigation, compared to the case when resilience is not allowed, reflected in 8c. Therefore mitigation spending when resilience is allowed is actually slightly higher compared to when it is not an option, resulting in comparable final climate risk in Figure 8b. This suggests that companies are financially incentivized to prioritize resilience investments, which can actually enable a greater commitment to climate mitigation efforts.

**Additional experiments.** In addition to the experiments presented above, we also explore settings where (1) company agents' actions are initialized with data from real-world companies, (2) agents' decisions are locked in for five years to simulate capital flexibility challenges, (3) companies' climate resilience parameter $L_t^{C_i}$ are random and vary across events and companies, and (4) the conditions for companies going bankrupt are made more strict. These experiments yield directional conclusions consistent with those presented in the main text. Detailed results can be found in Appendix 11.

## 6 DISCUSSION AND CONCLUSION

In this paper, we introduce InvestESG, a MARL environment simulating long-term interactions between companies and investors under varying ESG disclosure policies. **For policymakers**, InvestESG shows that mandatory ESG disclosure, paired with highly ESG-conscious investors, can drive corporate mitigation efforts. Additionally, greenwashing poses a lesser challenge than anticipated, and providing high-quality system-wide information effectively motivates action from both corporations and investors. **For economics and policy researchers**, InvestESG introduces MARL as a promising tool to complement traditional empirical and theoretical methods, allowing scalable policy testing in a simulated environment. Our model predicts agent behaviors consistent with empirical evidence and uncovers novel insights. **For the machine learning community**, InvestESG presents a novel multi-agent benchmark, fostering the development of RL algorithms that tackle complex social dilemmas, competition, and long-term strateg, and pushing forward AI applications in real-world, high-impact domains. We encourage the machine learning community to develop algorithmic innovations for InvestESG that can inspire practical actions to address climate change.

**Limitations.** We acknowledge that InvestESG simplifies various aspects of real-world climate evolution, financial markets, and regulatory frameworks. These simplifications were intentional for two reasons: (1) rather than fully simulating the global market and climate complexities, which is arguably infeasible for a single model, we aim to develop a "first-principles" model that offers directional insights for policy design; (2) we designed the benchmark to be simple enough for the ML community to iterate on with reasonable computational resources (e.g., a single GPU). By lowering computational barriers, we hope to encourage broader participation from the ML community.

At this stage, InvestESG effectively captures the core incentive structures necessary to evaluate the policy in question. We anticipate future improvements to include more complexity, allowing researchers and policymakers to select the appropriate level of granularity for their needs.

**Future Work.** Our goal is to spark discussions and evolve InvestESG into a robust tool for policy design, with input from experts in machine learning, climate change, and economics. We envision three key areas for future work: **(1) Richer environment.** For instance, we plan to incorporate employees and customers, who may also be attracted to more climate-friendly companies. We also envision capturing complex market and environmental dynamics between industries (e.g., include insurance companies (Brogi et al., 2022)), and enhancing investor decision-making to move beyond binary choices to better reflect real-world complexities. **(2) Enhanced policy design.** We propose testing policies that strengthen ESG disclosure mandates, such as guidelines on Scope 1 and 2 emissions, which, while costly, may ease the evaluation of companies' efforts. We also suggest exploring ways to redesign the simple ESG score to better inform investors and motivate corporate actions. **(3) Richer agent dynamics.** For instance, we plan to allow investors to learn ESG-consciousness level through interaction with the environment, which can change overtime. Given the challenges of passing new regulations, we aim to explore if companies and investors can self-regulate through coordination. For instance, we suggest testing if collective funds or peer monitoring can enhance mitigation or lead to collusion.

## 7 ETHICS STATEMENT

This research is directly aimed at improving human well-being by addressing the existential threat posed by climate change, one of the most pressing issues of our time. Our work seeks to develop a fully reproducible, open-source multi-agent reinforcement learning (MARL) benchmark, enabling the highest standard of transparency and scientific rigor. The benchmark is designed to be freely accessible to all researchers, and performant enough to run efficiently on commonly available GPU resources, fostering inclusivity. By simulating real-world social dilemmas between corporations and investors, we aim to spark novel solutions for reducing emissions and mitigating climate change risks. No human subjects are involved in this research.

## 8 REPRODUCIBILITY STATEMENT

To promote reproducibility, we provide our source code and corresponding configuration files used for running the experiments in open source. The details about installation and sample code for running the experiment are also included at `https://github.com/yuanjiayiy/Invest ESG`.

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

In the Appendix, we will cover the following details of our work.

- **Technical details of the InvestESG environment** Appendix 9: Math formulation of the environment.

- **Implementation details** Appendix 10: Implementation details about multi-agent reinforcement learning experiments.

- **Additional experiments** Appendix 11: Details about additional experiments, including the results with more agents in 11.1, the impacts of resilience spending on the stakeholders' behavior and mitigation level in **??**, the effects of seeding companies with real-world date in 11.2, the effects of lock-in period for agent decisions in 11.3, the effects of uncertain climate event damage in 11.4, and the effects of bankruptcy mechanism in 11.5.

## 9 TECHNICAL DETAILS OF THE INVESTESG ENVIRONMENT

The simulation starts in 2021 and runs through 2120, with each period $t$ corresponding to one year. The environment includes two main components: (1) an evolving climate and economic system, and (2) two types of agents: $M$ company agents $\mathcal{C}_i$ for $i \in \{1, \ldots, M\}$ and $N$ investor agents $\mathcal{I}_j$ for $j \in \{1, \ldots, N\}$.

**Climate and Economic Dynamics.** The environment is characterized by three climate risk parameters: extreme heat probability ($P_t^h$), heavy precipitation probability ($P_t^p$), and drought probability ($P_t^d$) in year $t$. Initial climate risks are set to $P_0^h = 0.28$, $P_0^p = 0.13$, and $P_0^d = 0.17$ following the IPCC estimate (Masson-Delmotte et al., 2021), resulting in an overall climate risk of $P_0 = 0.48$, the probability of at least one adverse climate event in a year. Without mitigation efforts, these risks increase linearly over time, reaching the IPCC's $4\,°C$ scenario scenario by 2100, which corresponds to the 80th period in our 100-period simulation. By that point, we observe $\overline{P}_{80}^h = 0.94$, $\overline{P}_{80}^p = 0.27$, and $\overline{P}_{80}^d = 0.41$ (or an overall climate risk of $\overline{P}_{80} = 0.97$). We then extrapolate this trend through to period 100. Figure 2a depicts how increased climate risks and adverse climate events increase over time in a scenario where companies are solely profit-motivated. Company agents can mitigate the growth of climate risk by investing in emissions reduction. The change in climate risk $P_t^e$ for event $e \in \{h, p, d\}$ in year $t$ is governed by the function:

$$P_t^e = \frac{\mu_e t}{1 + \lambda_e U_{t,m}} + P_0^e, \quad \text{for } e \in \{h, p, d\}, \tag{1}$$

where $U_{t,m}$ is the cumulative mitigation spending from all agents by period $t$. If $U_{t,m} = 0$, risks increase linearly to reach $\overline{P}_{80}^e$ by 2100 as explained earlier. When $U_{t,m} > 0$, the growth rate of climate risk decreases. The parameters $\lambda_e$ are calibrated based on Shukla et al. (2022), which estimates that an annual mitigation investment of \$2.3 trillion is required to achieve IPCC's $1.5\,°C$ scenario. The model fits $\lambda_e$ so that such investment levels would yield $1.5\,°C$ scenario climate risks by 2100. Climate events are modeled as independent Bernoulli processes, allowing for multiple events within a year (red dashed lines in Figure 2a). Let $X_{t,h}, X_{t,p}, X_{t,e} \in \{0, 1\}$ represent the occurrence of each climate event, and $X_t$ denote the total number of events in period $t$, determined by Equation 2.

$$X_t = X_{t,h} + X_{t,p} + X_{t,e}, \quad \text{where } X_{t,e} \sim \text{Bernoulli}(P_t^e), \quad \text{for } e \in \{h, p, d\}. \tag{2}$$

In addition to the evolving climate risks, the environment incorporates a baseline *economic growth rate* $\gamma$, set to 10% by default, aligned with the historical average annual return of the S&P 500 over the past century (Damodaran, 2024a). Company agents' capital levels $K_t^{\mathcal{C}_i}$ grow at rate $\gamma$ each year, barring climate events. If an adverse event occurs, company agents lose a portion of their total capital according to their respective *climate resilience* parameter $L_t^{\mathcal{C}_i}$. If economic losses drive a company's remaining capital into negative territory, the company is declared bankrupt.

**Company Action Space.** Each company agent $\mathcal{C}_i$ in period $t$ selects actions from a continuous vector $\mathbf{u}_t^{\mathcal{C}_i} = (u_{t,m}^{\mathcal{C}_i}, u_{t,g}^{\mathcal{C}_i}, u_{t,r}^{\mathcal{C}_i})$, where $u_{t,m}^{\mathcal{C}_i}$ represents the share of capital allocated to mitigation, $u_{t,g}^{\mathcal{C}_i}$ to greenwashing, and $u_{t,r}^{\mathcal{C}_i}$ to building climate resilience. The action space for each company is defined as a continuous 3-dimensional unit cube, $\mathcal{U}_t^{\mathcal{C}_i} = [0, 1]^3$. If the sum of the three ratios exceeds 1

in any period, the company agent is deemed to be overspending, resulting in bankruptcy. Mitigation spending directly reduces the system-wide climate risk as explained above. Greenwashing involves deceptive marketing or accounting tactics that allow companies to appear climate-friendly at a low cost without providing any real benefits to society (de Freitas Netto et al., 2020; Yang et al., 2020b). Resilience spending enhances the company's climate resilience by lowering its vulnerability $L_t^{\mathcal{C}_i}$, but it does not reduce emissions and therefore does not mitigate system-wide climate risk. In the default setting, we disable greenwashing and resilience spending to focus on testing companies' mitigation efforts. These options are later enabled to examine their effects, as will become clear in Section 5.

**Investor Action Space.** Investor agents first select the companies they want to invest in. Specifically, agent $\mathcal{I}_j$ in period $t$ selects an action from a binary vector of length $M$, $\mathbf{a}_t^{\mathcal{I}_j} = (a_{t,1}^{\mathcal{I}_j}, \ldots, a_{t,M}^{\mathcal{I}_j})$, corresponding to the $M$ company agents. Each entry $a_{t,i}^{\mathcal{I}_j} = 1$ indicates that investor $\mathcal{I}_j$ invests in company $\mathcal{C}_i$ in period $t$, and $a_{t,i}^{\mathcal{I}_j} = 0$ otherwise. The investor's action space at time $t$ is thus $\mathcal{A}_t^{\mathcal{I}_j} = \{0,1\}^M$. Once the choices are made, investors capitals are distributed equally among these chosen companies, as will be detailed later in Equation 4.

**Modeling ESG Disclosure.** With the ESG disclosure mandate in place, each company agent receives an updated ESG score $Q_{t+1}^{\mathcal{C}_i}$ in period $t$, calculated as

$$Q_{t+1}^{\mathcal{C}_i} = u_{t,m}^{\mathcal{C}_i} + \beta u_{t,g}^{\mathcal{C}_i}, \tag{3}$$

where $\beta > 1$ indicates that greenwashing is cheaper than genuine mitigation in terms of building an ESG-friendly image. This mirrors simple ESG ratings provided by some agencies, which typically range from 1 to 100 (S&P Global, 2024) or use letter-based ratings (MSCI Inc., 2024).

**State and Observation Space.** The environment simulates a partially observable Markov game $\mathcal{M}$ defined over a continuous, multi-dimensional state space. The system state at period $t$ is characterized by the three climate risk parameters $\mathbf{P}_t = (P_t^h, P_t^p, P_t^d)$, each company agent's state vector $\mathbf{S}_t^{\mathcal{C}_i} = (K_t^{\mathcal{C}_i}, Q_t^{\mathcal{C}_i}, L_t^{\mathcal{C}_i})$, where $K_t^{\mathcal{C}_i}$ is the capital level, $Q_t^{\mathcal{C}_i}$ is the ESG score, and $L_t^{\mathcal{C}_i}$ is the climate resilience. Each investor agent's state is represented by their investment portfolio and cash levels $\mathbf{S}_t^{\mathcal{I}_j} = (H_{t,1}^{\mathcal{I}_j}, \ldots, H_{t,M}^{\mathcal{I}_j}, C_t^{\mathcal{I}_j})$, where $H_{t,i}^{\mathcal{I}_j}$ represents investor $\mathcal{I}_j$'s holdings in company $\mathcal{C}_i$, and $C_t^{\mathcal{I}_j}$ is the investor's cash level. The full system state at period $t$ is thus $\mathcal{S}_t = \left( \mathbf{P}_t, \{\mathbf{S}_t^{\mathcal{C}_i}\}_{i=1}^M, \{\mathbf{S}_t^{\mathcal{I}_j}\}_{j=1}^N \right)$. All company and investor agents share a common observation space, denoted as $\mathcal{O}_t$. In the default setting, $\mathcal{O}_t = \left( \{\mathbf{S}_t^{\mathcal{C}_i}\}_{i=1}^M, \{\mathbf{S}_t^{\mathcal{I}_j}\}_{j=1}^N \right)$. Extensions that incorporate additional observable information, like climate risk, are explored in Section 5.

**State Transition.** The environment's state transition $\mathcal{T}$ proceeds as follows. At the beginning of period $t$, investors collect their investment holdings from period $t-1$ and redistribute their capital according to $\mathbf{a}_t^{\mathcal{I}_j}$. Denote $||a_t||_1^{\mathcal{I}_j} = \sum_{i=1}^M a_{t,i}^{\mathcal{I}_j}$ as the number of companies investor $\mathcal{I}_j$ invests in during period $t$ and let $K_t^{\mathcal{I}_j} = \sum_{i=1}^M H_{t,i}^{\mathcal{I}_j} + C_t^{\mathcal{I}_j}$ represent the total capital of investor $\mathcal{I}_j$ at the start of period $t$. Companies reach an interim capital level after returning old investments, $\sum_{j=1}^N H_{t,i}^{\mathcal{I}_j}$, and receiving new ones, with the investment from investor $\mathcal{I}_j$ calculated as $a_{t,i}^{\mathcal{I}_j} \frac{K_t^{\mathcal{I}_j}}{||a_t||_1^{\mathcal{I}_j}}$ or 0 if the investor opts out of investing, as shown in Equation 4.

$$K_{t+1,interim}^{\mathcal{C}_i} = K_t^{\mathcal{C}_i} - \sum_{j=1}^N H_{t,i}^{\mathcal{I}_j} + \sum_{j=1}^N a_{t,i}^{\mathcal{I}_j} \frac{K_t^{\mathcal{I}_j}}{||a_t||_1^{\mathcal{I}_j}}, \quad \text{for } i = 1, \ldots, M \tag{4}$$

Companies then make climate-related spending using the interim capital, as described in Equations 5 to 6. Here, $U_{t,m}$ represents the cumulative mitigation spending by all company agents up to period $t$, while $U_{t,r}^{\mathcal{C}_i}$ denotes the cumulative resilience spending by company $\mathcal{C}_i$.

$$U_{t,m} = U_{t-1,m} + \sum_{i=1}^M u_{t,m}^{\mathcal{C}_i} \times K_{t+1,interim}^{\mathcal{C}_i} \tag{5}$$

$$U_{t,r}^{\mathcal{C}_i} = U_{t-1,r}^{\mathcal{C}_i} + u_{t,r}^{\mathcal{C}_i} \times K_{t+1,interim}^{\mathcal{C}_i} \quad \text{for } i = 1, \ldots, M \tag{6}$$

While $U_{t,m}$ is then plugged into Equation 1 where system climate risks are updated. Equation 7 states that a company's climate resilience, $L_t^{\mathcal{C}_i}$, scales with the proportion of cumulative resilience investment relative to its capital, and that increasing resilience becomes progressively more challenging due to diminishing returns (Pörtner et al., 2022).

$$L_t^{\mathcal{C}_i} = L_0^{\mathcal{C}_i} \exp\left(-\eta^{\mathcal{C}_i} \frac{U_{t-1,r}^{\mathcal{C}_i} + u_{t,r}^{\mathcal{C}_i} K_{t+1,interim}^{\mathcal{C}_i}}{K_{t+1,interim}^{\mathcal{C}_i}}\right), \quad \text{for } i = 1, \ldots, M. \tag{7}$$

At the same time, the occurrence of climate events are simulated according to Equation 2, and companies receive updated ESG scores based on Equation 3. Afterwards, companies' profit margins, $\rho_t^{\mathcal{C}_i}$, are computed using Equation 8, factoring in climate-related spendings $u_{t,m}^{\mathcal{C}_i}, u_{t,g}^{\mathcal{C}_i}, u_{t,r}^{\mathcal{C}_i}$, default economic growth $\gamma$, and losses due to climate events, which are influenced by companies' climate vulnerability $L_t^{\mathcal{C}_i}$ and the number of climate events $X_t$ as defined in Equation 2:

$$\rho_t^{\mathcal{C}_i} = (1 - u_{t,m}^{\mathcal{C}_i} - u_{t,g}^{\mathcal{C}_i} - u_{t,r}^{\mathcal{C}_i})(1 + \gamma)(1 - X_t L_t^{\mathcal{C}_i}) - 1, \quad \text{for } i = 1, \ldots, M. \tag{8}$$

Company capitals $K_{t+1}^{\mathcal{C}_i}$, investor holdings $H_{t+1,i}^{\mathcal{I}_j}$, and investor cash positions $C_{t+1}^{\mathcal{I}_j}$ are updated according to Equations 9 to 11. Company capitals $K_{t+1}^{\mathcal{C}_i}$ are updated according to Equation 9 by scaling the interim capital levels by profit margin.

$$K_{t+1}^{\mathcal{C}_i} = (1 + \rho_t^{\mathcal{C}_i}) K_{t+1,interim}^{\mathcal{C}_i}, \quad \text{for } i = 1, \ldots, M. \tag{9}$$

Equation 10 adjusts investor holdings based on company profit margins in their portfolios.

$$H_{t+1,i}^{\mathcal{I}_j} = a_{t,i}^{\mathcal{I}_j}(1 + \rho_t^{\mathcal{C}_i}) \frac{K_t^{\mathcal{I}_j}}{||a_t||_1^{\mathcal{I}_j}}, \quad \text{for } i = 1, \ldots, M, j = 1, \ldots, N. \tag{10}$$

If an investor chooses not to invest, all capital remains as cash, as shown in Equation 11

$$C_{t+1}^{\mathcal{I}_j} = \begin{cases} 0, & \text{if } ||a_t||_1^{\mathcal{I}_j} \neq 0 \\ K_t^{\mathcal{I}_j}, & \text{if } ||a_t||_1^{\mathcal{I}_j} = 0 \end{cases}, \quad \text{for } j = 1, \ldots, N. \tag{11}$$

**Rewards.** The single-period reward for company $\mathcal{C}_i$ is solely based on its profit margin, given by $r_t^{\mathcal{C}_i} = K_{t+1}^{\mathcal{C}_i} - K_{t+1,interim}^{\mathcal{C}_i}$ for $i = 1, \ldots, M$, reflecting the assumption that companies are profit-driven. The reward for investor $\mathcal{I}_j$ is $r_t^{\mathcal{I}_j} = \frac{K_{t+1}^{\mathcal{I}_j} - K_t^{\mathcal{I}_j}}{K_t^{\mathcal{I}_j}} + \alpha^{\mathcal{I}_j} \frac{\sum_{i=1}^{M} H_{t+1,i}^{\mathcal{I}_j} Q_{t+1}^{\mathcal{C}_i}}{\sum_{i=1}^{M} K_{t+1}^{\mathcal{I}_j}}$ for $j = 1, \ldots, N$. The first component is the portfolio return ratio, and the second component represents the weighted average ESG score of the investor's portfolio adjusted by the investor's ESG preference, $\alpha^{\mathcal{I}_j}$.

**Social Outcome Metrics.** We evaluate agent performance based on two key social outcome metrics: the final climate risk level, $P_{100}$, defined as $P_{100} = 1 - (1 - P_{100}^h)(1 - P_{100}^p)(1 - P_{100}^d)$, and the total market wealth at the end of the period, $W_{100}$, defined as $W_{100} = \sum_{i=1}^{M} K_{100}^{\mathcal{C}_i} + \sum_{j=1}^{N} K_{100}^{\mathcal{I}_j}$.

## 10 IMPLEMENTATION DETAILS

### 10.1 INDEPENDENT-PPO

To test how self-interest agents learn to respond to incentives in the environment, we employ a state-of-the-art MARL algorithm based on Independent PPO. Each agent has its own policy parameters, and agents do not share parameters among themselves. This is because we are interested in simulating companies and investors as independent, selfishly motivated agents that specialize in maximizing their own expected reward. The policy model is a simple Multi-Layer Perceptron (MLP) network, with input as the capital, resilience and margin of each company, along with the investments and capital of each investor. Additional information can be added to the observation as in 4.2. All company and investor agents share a common observation space. To implement Independent-PPO with different roles, we built upon the Stable Baseline 3 repository (Raffin et al., 2021). Each policy model is an MLP network with two layers of size 256 and 128 and with tanh activation layers, and update each policy after 5 episodes during the training. See Table 1 for more details.

In the default setting, we assign equal amount of initial capitals to companies and investors, which is a rough representation of the current market. For each experimental scenario, we run the learning algorithms for 30k episodes, over 3 trials with different random seeds.

| MLP layers | Activation layers | PPO n_steps | PPO learning rate | PPO entropy coefficient | Gradient Clipping |
|---|---|---|---|---|---|
| 256, 128 | tanh | 500 | $3 \times 10^{-5}$ | 0.01 | 0.2 |

Table 1: IPPO policy training parameters. The rest of the parameters are the default as described in Stable Baselines 3 (Raffin et al., 2021).

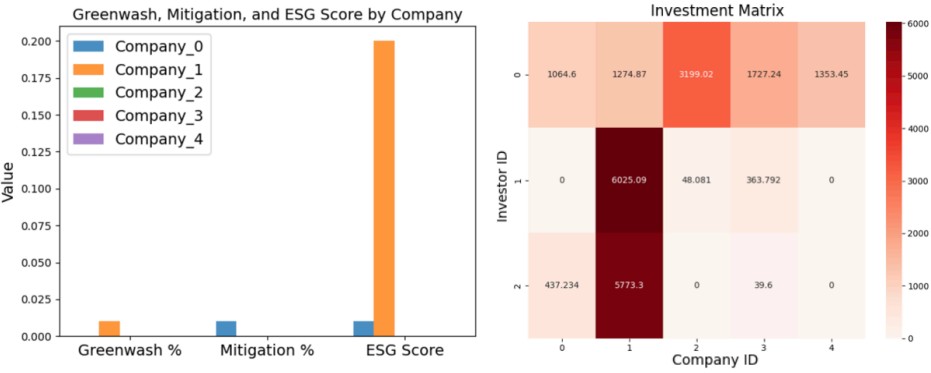

(a) ESG-conscious investors can be distracted by greenwashing.

Figure 9: ESG-conscious investors can be distracted by greenwashing, and heavily invest in a company that greenwashes. Here we examine a scenario where Company 0 is hard-coded to invest in real mitigation while Company 1 only invests in greenwashing. Investors have ESG consciousness level of 0, 1, 10.

## 11 ADDITIONAL EXPERIMENTS

### 11.1 SCALE UP THE NUMBER OF AGENTS

Figure 10a - 10b show the experiment results when the number of company and investor agents are scaled up to 25-by-25. The increased number of agents reveal the same directional story as the main results shown in Figure 4, where highly ESG-conscious investors motivate mitigation efforts from companies, resulting in lower ending climate risk and higher total market wealth. Figure 10c-10d zoom into a single episode of the 10-company, 10-investor case, with investor ESG-consciousness set to 0 and 10, respectively. In the ideal case where all investors are highly ESG-conscious, one leading mitigating company attracts the majority of the investment, which is consistent with the pattern shown in Figure 6a.

### 11.2 SEEDING COMPANIES WITH REAL-WORLD DATA

To ground the agents in real-world data, we seeded company agents' actions with real-world corporate behaviors. According to Gardiner & Associates (2023); European Investment Bank (2023); Partners (2024), approximately 50% of large companies are currently investing in climate mitigation. Globally, about 1% of GDP is allocated to climate finance annually (Buchner, 2023). Since GDP can be roughly viewed as the counterpart of a company's sales, and based on data from an NYU database (Damodaran, 2024b), which estimates the average sales-to-capital ratio across sectors to be between 0.8 and 1.28, we approximate sales and capital to be of similar magnitudes. Consequently, we seeded 50% of companies to invest between 0.5% and 1% of their total capital into mitigation, reflecting real-world investment levels. As shown in Figures 11a, 11b, and 11c, when seeded with real data, the total corporate mitigation efforts and resulting climate risk levels eventually align closely with the baseline scenario.

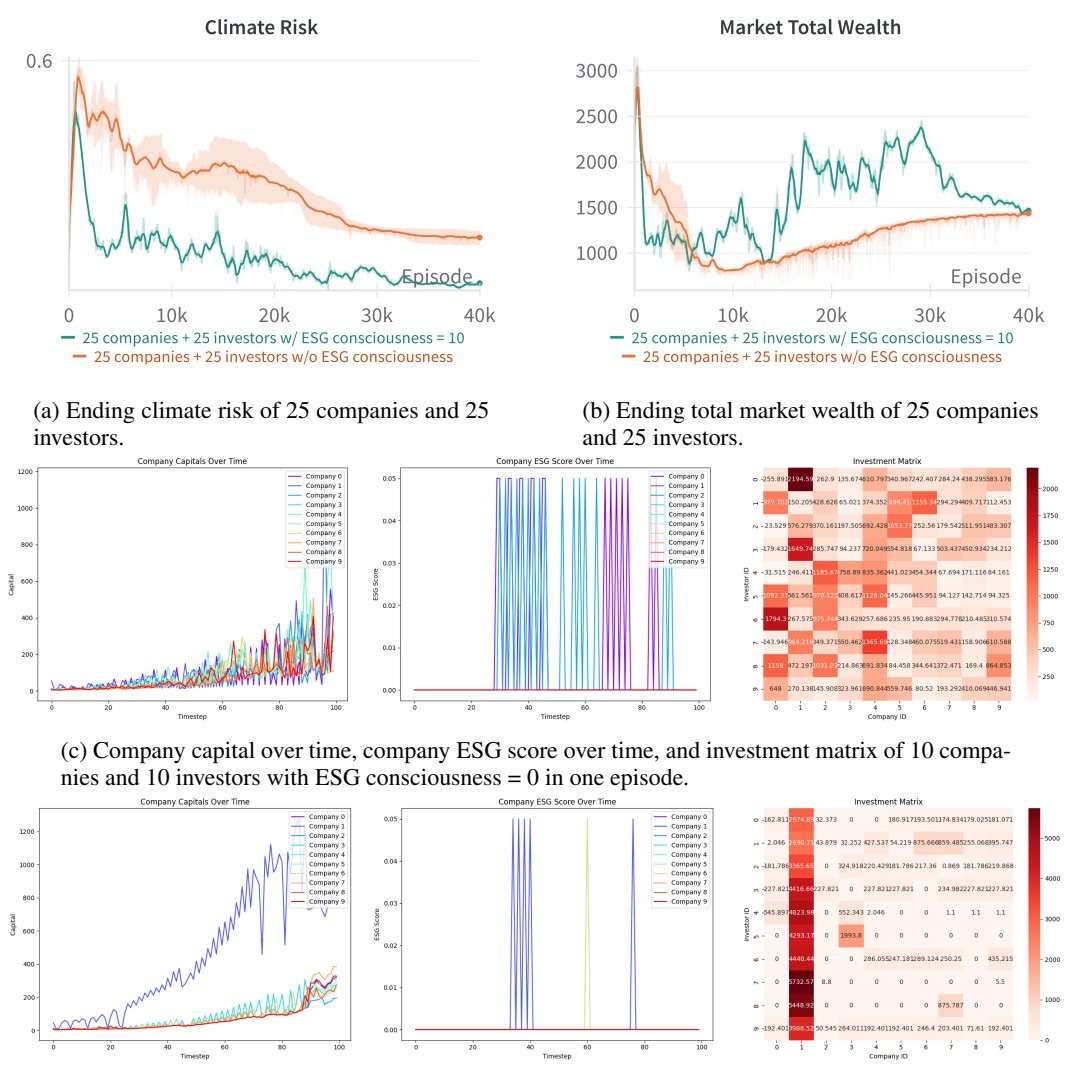

(a) Ending climate risk of 25 companies and 25 investors.

(b) Ending total market wealth of 25 companies and 25 investors.

(c) Company capital over time, company ESG score over time, and investment matrix of 10 companies and 10 investors with ESG consciousness = 0 in one episode.

(d) Company capital over time, company ESG score over time, and investment matrix of 10 companies and 10 investors with ESG consciousness = 10 in one episode.

Figure 10: (a)(b) shows the final climate risk and market total wealth for the case of 25 companies and 25 investors. Similar to the default 5-company-and-3-investor case, when investors are highly conscious, the final climate risk would be decreased. (c)(d) shows the ending episode company capitals over time, company ESG score over time, and investment matrix of environments with 10 companies. When investors are ESG conscious, the investments are more concentrated on the mitigating company compared to the case when the investors are not ESG conscious. Consequently, the capitals of mitigating companies are much larger compared to non-mitigating companies in the case when the investors are ESG conscious, while the capitals of mitigating companies are comparable to non-mitigating companies when the investors are not ESG conscious.

## 11.3 LOCK-IN INVESTMENTS

In reality, the decision-making processes of both companies and investors can be less flexible than modeled, where companies and investors update their strategies annually. To reflect the capital inflexibility, we implemented a 5-year lock-in period for agent decisions. This approach allows the climate to evolve more rapidly than agents can respond. Despite this constraint, the results remain consistent with the directional findings presented in the main text, as illustrated in Figures 11d, 11e and 11f, suggesting that agents would achieve similar results using a macro-action reinforcement learning approach (Durugkar et al., 2016). However, the learning curves for the locked-in investment

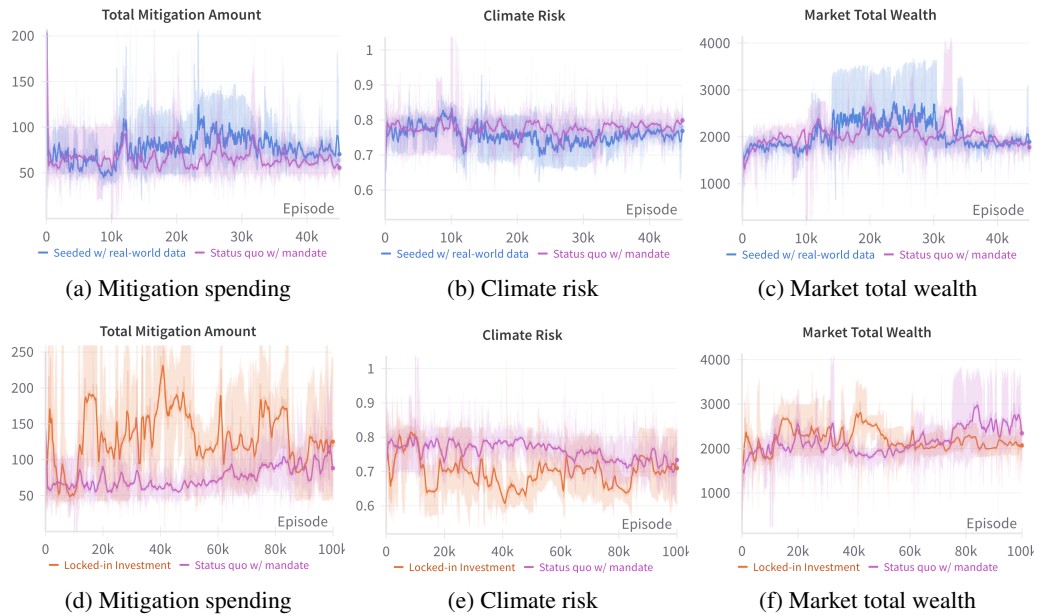

Figure 11: (a)-(c) Effect of seeding with real-world data. (d)-(f) Effect of 5-year lock-in period for agent decisions. All comparisons are made against the default case with an ESG disclosure mandate, in which investors have zero ESG-consciousness, and company actions are restricted to mitigation efforts only.

cases exhibit greater volatility and slower convergence compared to the default case, indicating that capital inflexibility indeed increases the challenges of addressing climate change.

## 11.4 UNCERTAIN CLIMATE EVENT DAMAGE

Given the significant uncertainty surrounding the economic damages of climate change (Farmer et al., 2015), we conducted additional experiments where companies' resilience parameters, $L_t^{C_i}$, representing economic losses from extreme climate events, were modeled as Gaussian random variables $L_t^{C_i} \sim \mathcal{N}(\mu, \sigma)$ clipped within the range [0,1], varying across both events and companies. This randomness allows for extreme climate damages, including the potential bankruptcy of some companies, which could incentivize risk-averse strategies where companies engage in greater mitigation efforts. Conversely, the randomness complicates the ability of company agents to learn the long-term benefits of mitigation, potentially encouraging short-sighted behaviors or greenwashing.

Figures 12a to 12c illustrate the effects of uncertain climate event damage compared to the default mandate case, where no investors are ESG-conscious, and company actions are restricted to mitigation only. Both scenarios result in similar levels of total mitigation and climate risk. However, agents in the uncertain damage scenario require more time to learn the optimal mitigation level due to the increased uncertainty in the environment. Despite the similarities, the uncertain damage scenario results in significantly lower total market wealth due to potentially some high-tail losses in early years. When considering the wealth discrepancy, companies in the uncertain damage scenario ultimately adopt a more aggressive mitigation strategy relative to their capital. This behavior suggests that the high uncertainty may have prompted risk-averse strategies.

Uncertain climate event damage also incentivizes the companies to focus more on short-term benefits by attracting investments, compared to the case with fixed damage when the companies first explore greenwashing strategy and then settle down on doing little greenwashing. As shown in 12d to 12g, when threatened with uncertain climate event damage, companies increase their greenwashing amount to attract immediate investments from ESG-consciouns investors.

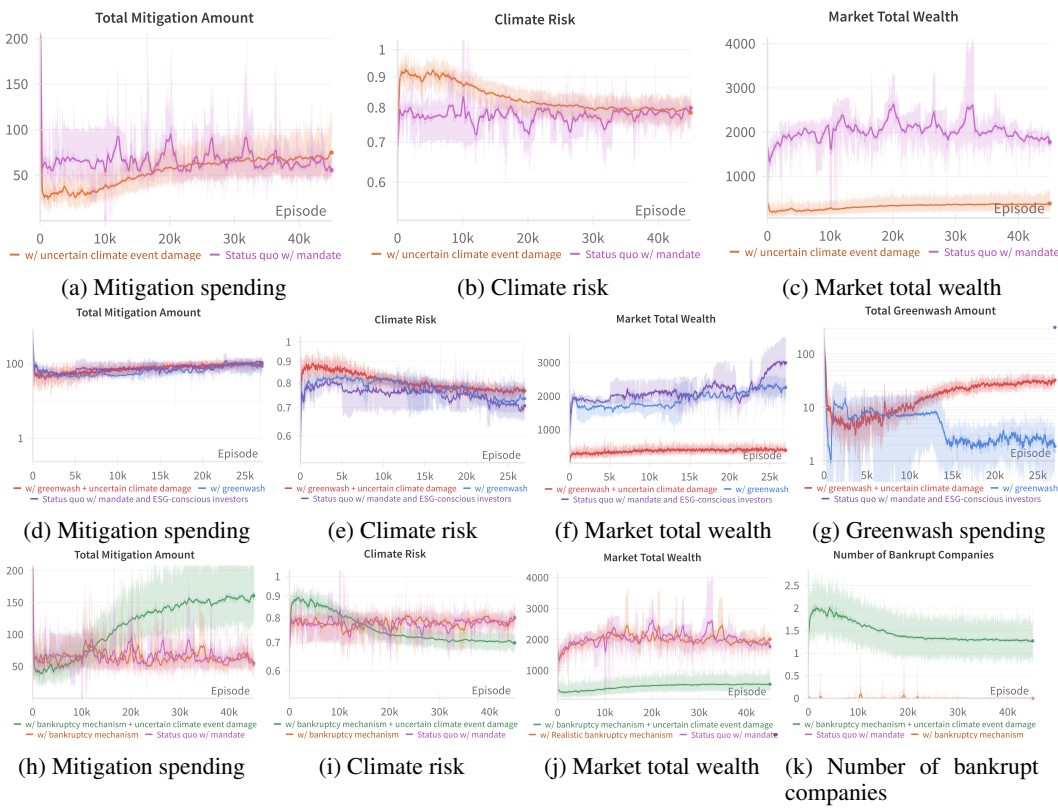

Figure 12: (a)-(c) Effect of uncertain climate event damage. (d)-(g) Effect of uncertain climate event damage on greenwash spending. (h)-(k) Effect of a more strick bankruptcy mechanism and its combination with uncertain climate event damage.

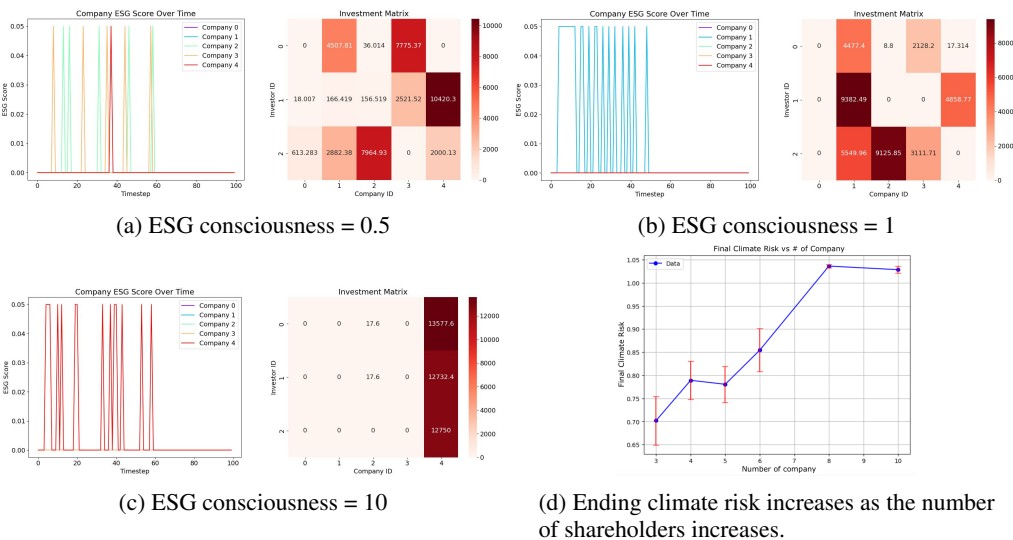

Figure 13: (a)-(c) shows the ending episode investment matrix for investors with different level of ESG consciousness. (d) shows ending climate risk vs. number of companies under the setting where no investors are ESG-conscious. When the number of companies increases, each company has less capital, and therefore companies have to spend a greater portion of their capital to see the same mitigation effect.

## 11.5 MORE STRICT BANKRUPTCY MECHANISM

Additionally, to make our model more enriched, we implemented a more strict bankruptcy mechanism for company agents: if a company agent has a margin worse than negative 10% for 3 consecutive years, a red flag signaling potential financial distress, is deemed as bankrupt (Shi & Li, 2019). As shown in the comparison between orange and pink lines in 12h, 12i and 12j, the new bankruptcy mechanism does not cause significant difference in the level of mitigation from the status quo with mandate case. Given that the market growth rate is high, companies are not severely threatened by bankruptcy despite that the new bankruptcy mechanism is in place, and therefore refrain from investing in climate mitigation.

However, when the stricter bankruptcy mechanism is combined with uncertain climate event damage, as depicted by the green lines in Figures 12h and 12k, a significant increase in mitigation and more bankruptcies are observed. This indicates that the combination of uncertain climate event damage and the bankruptcy mechanism creates an immediate risk of bankruptcy for company agents, thereby strongly incentivizing their mitigation efforts.

## 12 SUPPLEMENTAL FIGURES

Figure 13 shows a few supplemental plots which may be of interest. Figure 13a to 13c reveal how investor investments concentrate in mitigating companies as they become increasingly more ESG-conscious. In Figure 13d, we examine the impact of varying the number of companies under the status quo scenario with 5 companies and 3 investors who have zero ESG-consciousness, while maintaining an initial wealth distribution of 50-50 between companies and investors. In the absence of ESG-conscious investors, the ending system climate risk increases as the number of companies grows. This is likely because a higher number of companies results in each company having less capital, requiring them to allocate a larger proportion of their resources to achieve the same mitigation effect.

