# OpenReview forum: "InvestESG: A multi-agent reinforcement learning benchmark for studying climate investment as a social dilemma"
_ICLR.cc/2025/Conference — ICLR 2025 Poster_

### Official Review · Reviewer_fPY9 · 2024-11-04

**Soundness:** 4
**Presentation:** 3
**Contribution:** 2
**Rating:** 8
**Confidence:** 4

**Summary:**

The paper presents InvestESG, a MARL environment that studies the impact of ESG disclosures on company and investor agents. The benchmark is meant to simulate companies' investment decisions into climate mitigation, green washing and resilience spending as a social dilemma.

Specifically, the contributions are:

* InvestESG, a climate-economic environment in which investors fund companies, which make decisions about how much to invest in climate-related spending over 100 years starting in 2020.
  * Climate risks grow linearly in the absence of any mitigation
  * Companies decide how much to spend on mitigation, greenwashing and resilience
  * Investors decide which companies to invest in based on their preferences, which trade off between profits and climate efforts documented by ESG disclosures.
  * As the simulation proceeds, companies make profits which they return to investors, while climate risks grow, resulting in a higher probability of extreme events.
  * Agents are modelled using IPPO

* A set of experiments shedding light on agent behaviour in InvestESG
  * With no ESG disclosures, purely profit-driven decisions result in suboptimal collective outcomes
  * The impact of ESG disclosures depending on how many and how much investors care about ESG reports when choosing which companies to invest in
  * Whether companies leverage greenwashing when it is allowed in InvestESG
  * Whether visibility of the climate-related risk probabilities impacts agent behaviour

* Conclusions for policymakers and researchers
  * Mandatory EST disclosure paired with ESG-conscious investors can drive corporate mitigation efforts.
  * Knowledge of climate risks motivates investors and companies
  * Agent behaviour is consistent with empirical evidence
  * InvestESG is an example of using MARL to tackle complex social dilemmas in real-world, high-impact domains

**Strengths:**

# Originality

* **Novel MARL application to ESG disclosures**: Even though Zhang et al. explore MARL in the policy space, as far as I know, this is the only MARL simulator that looks at ESG disclosure impact in this scenario.

* **Novel problem formulation**: the authors cleanly describe the relationship between companies and investors with a two agent type system, as well as an ESG disclosure component.

# Quality

* **Relevant problem setup**: key decisions are captured by the problem setup. The ESG disclosure abstraction is simple and elegant. The reward structure effectively represents a social dilemma.

* **Extensive experimental results**: the authors go through many scenarios with InvestESG to analyze different outcomes.

# Clarity

* The paper is **well structured**, and makes for a smooth read with little to no cognitive breaks.

* The work is **well situated** within the literature on MARL simulators, and they contrast well with similar work.

* The design and implementation of InvestESG is **clearly laid out**.

* The work makes judicious use of **relevant visualizations**, such as Schelling diagrams.

# Significance

* The analysis is **timely and relevant** given the current discussions around ESG disclosures.

* The conclusions around the preferences of investors for climate-active companies is impactful.

* The use of MARL to study social dilemmas is an important subject of study.

**Weaknesses:**

# Soundness

* **The economic agents are not grounded in the economics literature**. This leads to issues such as capital being perfectly flexible across time steps. In traditional economics models, investments in capital last and they are not flexible. Here, there seems to be an implied assumption of perfectly flexible capital, which is unrealistic. Starting with an existing model of economic agents (with a citation), highlighting its limitations for InvestESG and then explaining how you extend to agent to accommodate for these limitations would be a much more compelling presentation.

* **Investor decisions are binary**, as opposed to continuous across all companies. Making investor decisions floats, i.e. a vector whose sum is capped at one, would allow for proportional investments across different companies. This is essential for investor diversification, which would also enable interesting extensions like regional damages to companies (i.e. climate events could affect subsets of agents either chosen at random or chosen somehow).

* Figure 7 b) is highly confusing. It looks like **with climate information, risk is *maximized* and market wealth is *minimized***. I'm not sure what exactly is going on in this plot, but it doesn't fit with the storyline of the paper. That is, it certainly does not look like more information improves decision making in this plot, if anything the effects of more information are catastrophic for both climate risk and market wealth.

* Figure 2b could be improved by showing the average number of events at each year across many episodes, as opposed to a single episode.

* The **number of agents is limited**. Granted, it is more than 2. However, it would be interesting to scale it up to more and see what types of behaviour emerge. There are group size effects that can emerge at scale in economics, e.g. see https://www.aeaweb.org/articles?id=10.1257/mic.20200290. This shows in section 9.2 of the paper in the appendix, but given the implications of such a result, it would be very important to expand upon these results.

# Presentation

* The paper is well structured, but the plots are a pain to read. The labels and ticks are too small, and the axes are not annotated.

* If you use a pdf format for your images instead of png, you can avoid the graininess when zooming in, which is necessary because of the label sizes.

* The results section could benefit from additional structure. It would be less dense and easier to read if you highlighted which of your results you consider the main results, and which you consider additional.

* I found the description of schelling diagrams fairly unclear, it took me a minute to get it.

* It should be ICLR 2025, not 2024. Please make sure the template you used is up to date.

* Inconsistent use of "MARL" and "multi-agent RL"

- Might benefit for a problem setting section, where you introduce important concepts like bifurcated equilibria

- Typo: 3.2 "self-interest" -> "self-interested"
# Contribution

* The importance of the contributions are weakened by what Figure 9 d) is suggesting, since there are many companies in the world. It seems to me that, without addressing the concerns raised by this result, your conclusions for policymakers do not hold.

**Questions:**

- What does a sensitivity analysis of the Beta parameter do to the results?

- How would more longer capital investment timelines (e.g. min 5 year lock-in) impact the trained agents?

- Do observations include past climate events?

- How did you calibrate the 0.5% investment of capital into mitigation?

- Why do you think that agents are so insensitive to the value of Beta as shown in figure 6 b)?

---

> ### Author Response · Authors · 2024-11-16
> **Response to Reviewer fPY9**
>
> Thank you for the very valuable and detailed feedback. We are grateful for your insights and the time you took to review our manuscript. Below, we address your comments.
>
> **Soundness**
> > The number of agents is limited.
> - We acknowledge that the number of agents in our experiment is limited compared to the real world. Post-submission we have developed a Jax version of the environment which allows us to run the simulations much faster and at a scaled level.  We will run larger-scale scenarios (e.g. 25 companies and 25 investors) to test the robustness of our findings. We believe that this number would match the expectation for a reasonably large group size as mentioned in the paper you suggested, unless you have alternative suggestions on the number of agents. We will share updates as soon as the new results are available. And we will also provide the Jax code in our updated version of the paper.
>
> > grounded in the economics literature.”
> - We really appreciate your suggestion on “Starting with an existing model of economic agents (with a citation), highlighting its limitations for InvestESG and then explaining how you extend to agents to accommodate for these limitations would be a much more compelling presentation.” As a quick response, we want to note the following reasoning. We would love your feedback on if this meets the need, and we can revise the paper to make the presentation stronger as you suggested.
>   - Our work is grounded in the economics and finance literature that studies similar questions with theoretical models. For example, in a recent highly impactful study published in a top finance journal, Pastor et al. (2021) built an analytical model for a single-period equilibrium that examines firm-investor tradeoffs much similar to ours. In their model, firms choose the “greenness” level, which affects their cost of capital, while investors select portfolios to maximize utility derived from both financial returns and their “taste” for green assets. Other influential studies in financial economics use comparable or even simpler frameworks. For example, Pedersen et al. (2021) model a single-period equilibrium with investors differentiated by their types of ESG preference, assuming fixed ESG characteristics for firms.
>   - Our simulations agree with the key findings in these papers suggesting that ESG-conscious investors prioritize green companies and promote positive social impacts by shifting investment towards green firms.
>   - Our framework, albeit still simple, allows for investigating more realistic and complex settings than the existing financial economics literature. These include (1) climate evolution, where companies mitigate emissions to attract investment and reduce long-term climate risk exposure—often omitted in financial studies (2) the potential for greenwashing, linked to information asymmetry (e.g., Lyon and Maxwell, 2011), and (3) a dynamic multi-agent game where firms and investors interact over many periods, which is more realistic. The equilibrium of such a setting is challenging to solve analytically or numerically (Pakes and McGuire, 2001), and our simulations provide insights here.
>   - Despite the simplicity of the framework, our simulation results match real-world data collected from countries that have implemented an ESG disclosure mandate of some sort as we discussed in section 4.
>
> > investments in capital last and they are not flexible;  longer capital investment timelines
> - Thank you for your insightful point regarding the investment timeline. To address this, we propose running an additional scenario in which agents can make investment decisions only every five periods, reflecting a five-year lock-in on their choices. Would this approach adequately address your concern?
>
> > Investor decisions are binary, as opposed to continuous across all companies.
> - We acknowledge that binary investor decisions simplify reality. However, to clarify, in our current setup, investors do not select only one company; they choose a set of companies to invest in each timestep, and distribute their current capital evenly to each of the selected companies, which allows for diversification within the binary framework. Further, because they can re-allocate investments at each timestep, they can achieve fine-grained diversification over the course of one episode. However, we agree that using continuous decision variables that sum to 1 would more accurately reflect real-world behavior, in which an investor does not always invest evenly. Could you please advise if you consider this a critical change that we should make during the rebuttal period? We are currently planning to prioritize experiments incorporating more agents, real-world data (based on reviewer mQaB’s suggestion), and the longer capital investment timelines experiment above.

---

> ### Author Response · Authors · 2024-11-16
> **Response to Reviewer fPY9**
>
> (cont.)
> > Figure 7 b) is highly confusing. It looks like with climate information, risk is maximized and market wealth is minimized.
> - Thank you so much for this catch. It turns out that it was an unintentional mistake from our side that we flipped the labels around. The actual result is that  with climate information, risk is minimized and market wealth is maximized. We will correct this mistake in our updated version of the manuscript.
>
> > Figure 2b could be improved by showing the average number of events each year across many episodes, as opposed to a single episode.
> - Thank you for the suggestion. We aimed to give an example of environmental dynamics and the progression of the environment in a single episode, and therefore we didn’t take the average of many episodes. However, based on your suggestion and reviewer mQaB’s suggestion to include more climate outcome metrics, we will provide additional plots of the average number of climate events across many episodes in the results for different scenarios.
>
> **Presentation and Questions**
> > Figures and plots
> - Thanks for the very constructive feedback regarding the presentation of the paper. We apologize for the inconvenience. We will increase the readability of our plots and fix the typo in the updated version accordingly.
>
> > results section could benefit from additional structure”
> - Thank you for your suggestion. Our intention was to tell the story in a progressive way,  starting from the baseline, then adding the ESG disclosure policy, followed by raising questions about investors with various level of environmental consciousness, and studying the impacts of greenwashing and sharing additional climate information. Following your feedback, we will restructure the result section by presenting the main story of baseline and behaviors of investors with different levels of consciousness, and delineate these central findings from the additional results from greenwashing and climate information. Would this satisfy your concerns? We will incorporate your feedback in the problem setting section in regard to introducing the important concepts.
>
> > What does a sensitivity analysis of the Beta parameter do to the results? Why do you think that agents are so insensitive to the value of Beta as shown in figure 6 b)?
> - We would like to use a sensitivity analysis of greenwashing coefficient β to elaborate our point that companies’ decision to abandon greenwashing isn’t due to the relative cost of greenwashing compared to real mitigation. Although companies that achieve a high ESG score through cheap greenwashing, which is controlled by β, attract more investments from investors, the increase in investment from investors by greenwashing isn’t comparable to their capital loss when exposed to severe climate events. The company agents learned greenwashing isn’t the most cost-effective way to maximize their capital no matter how much it costs.
>
> > Do observations include past climate events?
> - The observation doesn’t include the past climate events by default (in the baseline case), but it is configurable in the environmental setting. In figure 7, we studied the effects of having additional climate information on mitigation, which include the number of severe climate events that occurred in the past year.
>
> > How did you calibrate the 0.5% investment of capital into mitigation?
> - Since company actions are continuous, we needed to select a specific cooperation level to illustrate the social dilemma structure in the Schelling diagram. We chose 0.5% as a representative benchmark, aligning with Patagonia's aggressive cooperative approach, where approximately $100 million (around 0.3% of its $3 billion valuation) is allocated annually to climate initiatives. The 0.5% choice is thus in a similar order of magnitude, and the shape of the diagram remains consistent across other values we tested.
>
> Thank you again for your valuable feedback. We hope our responses address your concerns, and we welcome any further questions or comments you may have. Your insights are instrumental in helping us strengthen this work, and we look forward to your continued guidance during the rebuttal phase.
>
> Reference:
>
> [1] Pástor, Ľ., Stambaugh, R. F., & Taylor, L. A. (2021). Sustainable investing in equilibrium. Journal of financial economics, 142(2), 550-571.
>
> [2] Pedersen, L. H., Fitzgibbons, S., & Pomorski, L. (2021). Responsible investing: The ESG-efficient frontier. Journal of financial economics, 142(2), 572-597.
>
> [3] Lyon, T. P., & Maxwell, J. W. (2011). Greenwash: Corporate environmental disclosure under threat of audit. Journal of economics & management strategy, 20(1), 3-41.
>
> [4] Shek, Katia. (2023, January 30). *Patagonia: ESG's Golden Child.* https://insights.grcglobalgroup.com/patagonia-esgs-golden-child/

---

> ### Comment · Reviewer_fPY9 · 2024-11-19
>
> Thank you for your detailed response to my concerns.
>
> Below, I present the responses in the same order that the authors opted have opted for.
>
> ---
>
> > Limited number of agents
>
> Thank you for your response. Given that the closest related work mentioned ([1]) runs their version with 27 agents, 25 agents seems like more than enough. In particular, it will be of interest to see if the conclusions for policymakers are robust to this increase in the number of agents, considering the implications of figure 9 d). Also, section 9.1's reference to figure 9 d) discusses resilience spending, but the figure itself plots climate risk as a function of the number of agents.
>
> > grounded in economics literature
>
> Thank you for your response. I believe the inclusion of such a discussion in the paper would be more than sufficient.
>
> > flexible investments
>
> A 5-year lock-in period could certainly help simulate the lack of flexibility of capital.
>
> > binary investment decisions
>
> Thank you for your response. Should the number and diversity of agents increase (as suggested above), it might not be necessary to modify the action space, which could complicate the learning dynamics. I do not believe this is a critical change.
>
> > results section structure
>
> Thank you, this would more than satisfy my concerns.
>
> > Beta parameter sensitivity analysis
>
> Thank you for your response. This highlights for me a limitation of the proposed framework. The agents, by training over many episodes, essentially learn a representation of the unobservable ground truth damage climate-economic damage function that relates climate events to economic damage as a function of their decisions. However, in reality, there is significant uncertainty in the level of economic damages (e.g. see [this article, section 3.4](https://link.springer.com/article/10.1007/s10640-015-9965-2#:~:text=A%20third%20approach%20to%20understanding,Page%202007%3B%20Macal%20and%20North)). This clashes for me with the conclusion that companies are insensitive to greenwashing efforts, as the supposed uncertainty around climate change damages might lead certain companies to greenwashing to please investors, while simultaneously disregarding (knowingly or unknowingly) future impacts.
>
> ---
>
> Thank you again for your detailed response.
>
> An interesting direction of future work for this would be enabling investors to *learn* their own ESG consciousness levels over many episodes. If you consider a setup where the increases in probabilities of catastrophic events are somewhat stochastic, then investors could learn to use the history of catastrophic events to update their preference for ESG consciousness during a given episode.
>
> This brings up another important direction of future work: allowing for companies that go bankrupt. A different perspective on emissions is to look at them as a mostly-unmeasured negative externality vs the measured economic performance of different companies. If a company goes bankrupt in 2050 after emitting heavily for 50 years and employing no mitigation, then the mitigation effort must be undertaken by others to reach the optimal collective trajectory. Including the possibility of bankruptcy would improve the realism of the simulation, while also providing a new interesting motivation for greenwashing. I'd be interested to see what impact this would have on ESG-conscious investors. This could also include the cost of climate audits to overcome greenwashing, sharing of such information among investors, etc.
>
> Another interesting work would be to consider investors that have a preference for companies that have a history of ESG-conscious investors.
>
> Despite the unsound approach of learning a representation of the ground truth damage function over many episodes, I am conscious of the overwhelming diversity of design choices in such a situation. I believe this work is for the most part sound and heads in an **impactful, novel research direction**, paving the way for very interesting future research. Normally, I would raise my score to a 7, but this option is not available this year. Therefore, **I maintain my score** for now. However, I am happy to continue the discussion around now-raised ground truth limitation, should you believe this is not a caveat for the conclusions you draw, or should you have alternate solutions to propose. I am also amenable to additional results based on the Jax rewrite.
>
> I wish to reiterate the **novelty and impact** of this proposed submission. I look forward to seeing where future work on this project goes.
>
> [1] Zhang, Tianyu, et al. "AI for global climate cooperation: modeling global climate negotiations, agreements, and long-term cooperation in RICE-N." arXiv preprint arXiv:2208.07004 (2022).

---

> > ### Author Response · Authors · 2024-11-25
> >
> > Thank you for continuing to engage with us and sharing detailed feedback and exciting new ideas!
> >
> > >Limited number of agents, flexible investments, grounded in economics literature
> >
> > We have run the suggested changes and across all these experiments, we observed the same directional results as in our original experiments. Since the system does not permit us to share figures here, we will include these findings in the updated version of our paper, which will be uploaded by 11/27. We will also revise our paper according to your other suggestions, as well as incorporate the above discussion of how our work is grounded in economics literature.
> >
> > >The ground-truth limitation
> >
> > Thank you for raising the critical point regarding the “unobservable ground truth climate-economic damage function” and the “significant uncertainty in the level of economic damages.” We agree that there is substantial uncertainty in companies’ economic losses resulting from climate events. To address this limitation in our current deterministic damage function, we propose conducting an experiment where economic losses from climate events are modeled as random variables within the range [0,1]. These losses will vary both across events and between companies. We believe this approach addresses two concerns:
> > - __Uncertainty and Realism__: Our experiment more closely aligns with the real-world ambiguity surrounding the economic impacts of climate events. While agents may still learn the distribution of these damages over time, increasing the variance of the random variable can ensure that the uncertainty remains significant enough for agents to consider disregarding the learned impacts, as real-world companies may do, because it’s too noisy.
> >
> > - __Bankruptcy Dynamics__: To clarify, our current framework already forces companies to go bankrupt if their capital becomes negative. However, in the experiments conducted so far, bankruptcy has not occurred because companies maintain an underlying economic growth rate sufficient to offset climate-induced losses and have learned not to overspend on climate-related efforts. By introducing a highly stochastic damage function, climate events could become severe enough to trigger bankruptcies. This adjustment could incentivize behaviors such as greenwashing (“if bankruptcy is inevitable in certain scenarios, it may be rational to prioritize immediate economic survival over long-term climate commitments”). We will report results on the number and frequency of companies going bankrupt in our existing experiments and the new experiments in the updated paper as well.
> >
> > We understand that you may be concerned that RL agents “see into the future” by being trained over many episodes, whereas human beings cannot. It is true that fundamentally, RL works by learning to estimate future rewards over many experiences with the environment. Our rationale behind using RL agents as an approximation of rational actors is, although humans and companies cannot directly observe the future, they make decisions based on experience, reasoning, and expectations about how the world evolves, often with greater sophistication than RL agents trained from scratch over repeated episodes. RL agents mimic this process by iteratively learning from simulated episodes, which can be seen as analogous to the iterative learning process humans undergo through trial, error, and observation of historical patterns. For example, stock prices in financial markets reflect the aggregate expectations of rational agents regarding future economic conditions. Similarly, RL agents do not "know" the future; rather, they estimate expected outcomes by averaging over many plausible scenarios and optimizing their policies based on this probabilistic understanding.
> >
> > By introducing uncertainty into key elements such as the climate-economic damage function (as described earlier), we create an environment that requires RL agents to operate under significant ambiguity, mirroring real-world decision-making processes.
> > Moreover, we believe that using RL agents currently represents the most effective method for approximating intelligent, rational agents at scale—such as individuals and corporations—who respond dynamically to incentives in this environment. This approach provides a practical and scalable way to explore emergent behaviors and policy outcomes in environments where traditional analytical methods are insufficient.
> >
> > >Investors learn ESG consciousness
> >
> > Thank you for proposing this incredibly interesting idea! We could test this by making investors receive negative utility for extreme weather events. Then they could learn to invest in climate-conscious companies to reduce their own probability of undergoing extreme weather events. Although we may not be able to obtain these results before 11/27, we love this idea and will definitely explore it in ongoing work.

---

> > > ### Comment · Reviewer_fPY9 · 2024-12-03
> > >
> > > Thank you for following up on the proposed experiments and ideas. I apologize for the brevity of my response: as all concerns have been addressed, I have opted to raise my score to 8.

---

### Official Review · Reviewer_mQaB · 2024-11-05

**Soundness:** 3
**Presentation:** 3
**Contribution:** 2
**Rating:** 5
**Confidence:** 4

**Summary:**

This paper presents InvestESG, a novel multi-agent reinforcement learning (MARL) benchmark designed to simulate and analyse the impact of varying Environmental, Social, and Governance (ESG) disclosure policies through the social dilemma paradigm. InvestESG  uses two types of agents: companies and investors. Companies allocate capital across mitigation, greenwashing, and resilience, with varying strategies influencing climate outcomes and investor preferences. The findings are consistent with empirical research using real-world data. They capture the positive impact of companies using information about global climate risks to determine their level of investment in mitigation, even without investor involvement. The paper is beautifully written and rigorous.

**Strengths:**

The paper represents a novel contribution in a highly relevant, high-impact domain, at the intersection between climate change and MARL. It is beautifully written and self-contained, with rigorous specifications of the InvesESG environment. The implementation details and code are provided, and overall the paper makes a good case for a MARL benchmark for studying climate investment through the social dilemma paradigm, via two agent types: companies and investors. InvestESG is designed to simulate and analyse the impact of varying Environmental, Social, and Governance (ESG) disclosure policies on corporate climate investments. In InvestESG, companies allocate capital across mitigation, greenwashing, and resilience, with varying strategies influencing climate outcomes and investor preferences. The findings are consistent with empirical research using real-world data. The results capture the positive impact of companies using information about global climate risks to determine their level of investment in mitigation, even without investor involvement.

**Weaknesses:**

The main weakness of the paper consists in its simplifying assumptions, in terms of the types of agents, and the considered scenarios, analysis and discussions. These limitations are acknowledged in the paper. Due to these reasons, I believe that, in the current format, the paper makes an insufficient contribution for a top conference like ICLR.

For a more significant contribution, this work could be extended in one or more possible directions: extend the agent types (possibly consider insurance companies/market?), add more complex agent behavior, learn parameters and behaviors from real data, include more social outcome metrics (in addition to the final climate risk level and the final total market wealth, at the end of the simulation period) and/or include additional features, such as agent bankruptcy, and a dynamic number of agents.

Assuming the agent-types remain just companies and investors, increasing the number of companies and investors, and learning their behavior from real world data, may be a sufficient extension for a more significant contribution.

In the longer term, the initial vision of InvestESG would benefit from a more diverse agent space, for a more realistic climate-change problem specification (however, this is not essential for a significant contribution).

**Questions:**

I think the paper could be extended in several possible directions, as indicated in the Weaknesses section, for a more significant contribution. Another possible direction would be to implement and assess the impact of other PPO policies than IPPO on the overall behaviour and insights.

Potential relevant papers are suggested below.

Bisaro, Alexander, and Jochen Hinkel. "Governance of social dilemmas in climate change adaptation." Nature Climate Change 6, no. 4 (2016): 354-359.

Bettini, Matteo, Amanda Prorok, and Vincent Moens. "Benchmarl: Benchmarking multi-agent reinforcement learning." Journal of Machine Learning Research 25, no. 217 (2024): 1-10.

Bettini, Matteo, Ryan Kortvelesy, and Amanda Prorok. "Controlling Behavioral Diversity in Multi-Agent Reinforcement Learning." arXiv preprint arXiv:2405.15054 (2024).

Brogi, Marina, Antonella Cappiello, Valentina Lagasio, and Fabrizio Santoboni. "Determinants of insurance companies' environmental, social, and governance awareness." Corporate Social Responsibility and Environmental Management 29, no. 5 (2022): 1357-1369.

---

> ### Author Response · Authors · 2024-11-16
> **Response to Reviewer mQaB**
>
> Thank you for your insightful comments on our paper. We will work on incorporating your comments in our paper to increase the significance of our contributions. We would like to address some of your comments below.
>
> **Suggested literature**: We appreciate the list of literature you provided, which covers both MARL benchmarking papers from the ML community and empirical analysis of the climate social dilemma and climate ESG awareness of insurance companies from the climate community. We will cite these references in our paper update and use them to guide our future work.
>
> **Proposed new experiments to address the detailed concerns**: We appreciate the point of creating a richer environment. We propose to address your concerns by running some new scenarios as listed below. Please kindly let us know if these scenarios sufficiently address your feedback, or if there are additional analyses you would recommend.
> > increasing the number of companies and investors
>   - Thank you for the suggestion on running more agents. We agree with this. Post-submission we have developed a Jax version of the environment which allows for scalable and performant simulations.  We will run larger-scale scenarios with more agents (eg. 25 companies + 25 investors) to test the robustness of our findings. We will share updates as soon as the new results are available. And we will also provide the Jax code in our updated version of the paper.
>
> > learn parameters and behaviors from real data
> - To our knowledge, there isn’t a comprehensive database that currently tracks company behaviors in a way that can be readily analyzed, especially since the EU's ESG disclosure mandate only launched this year and the U.S. mandate is still pending. Therefore, we propose to use public information to estimate (1) the percentage of companies that invest in climate mitigation; and (2) for those companies, the amount of capital they allocate to mitigation.
>   - According to public sources (Miller, 2023; European Investment Bank, 2023; Climate Impact Partners, 2024), about 50% of large companies are investing in mitigation.
>   - We propose a range of 0.1\~1% of capital allocated to climate mitigation based on the following reasoning.
>     - As noted earlier, there is no readily available database that tracks companies' actual climate spending. For instance, while the UK is one of the few large economies to implement a disclosure mandate, it does not require companies to report specific climate expenditures. Moreover, fewer than 5% of companies voluntarily disclose such information (LSEG, 2023).
>     - Patagonia spends \~0.3% of its capital annually on climate initiatives (GRC Insights, 2023).
>     - The top five oil companies allocated 12% of CAPEX to low carbon activities annually, which translates to about 1.2% of their capital (InfluenceMap, 2022).
>     - About 1% of global GDP goes to climate finance, which includes public spending (Climate Policy Initiative, 2023). We can view GDP as roughly the counterpart of a company's sales. According to an NYU database, the average sales/capital ratio across sectors is 0.8~1.28 (Damodaran, 2024), i.e. on average sales is on par with capital in terms of order of magnitude. So we can also seed with 1%, although this is likely an upper bound as public spending tends to be higher than private spending in this domain.
>
>   - We will run a simulation where for the first 5 years 50% of companies spend 0.1\~1% of capital on mitigation. We randomly select which 50% but keep it consistent. Could you please advise if this suggested experiment would meet your criteria?
>
> > include more social outcome metrics (in addition to the final climate risk level and the final total market wealth, at the end of the simulation period)
> - We can add more social outcome metrics as you mentioned, such as the total number of severe climate events, or financial losses due to climate events during a certain period. Please let us know if there are other specific metrics you would recommend.
>
> > include additional features, such as agent bankruptcy
> - To clarify, we do allow agents to go bankrupt if their capital turns negative. When a company goes bankrupt, it is blocked from further actions, and investors holding equity in the company lose their investment. This is briefly mentioned in Section 3 *Company Action Space*, and we will make it clearer in the updated version.

---

> ### Author Response · Authors · 2024-11-16
> **Response to Reviewer mQaB**
>
> (cont.)
> > Extend the agent types (possibly consider insurance companies/market?), add more complex agent behavior
>
>  - Thank you for suggesting insurance companies as an additional agent type. As noted in Section 5 (Future Work), we plan to continue enriching the environment, and we will include insurance companies in our future developments, as they play a critical role in climate-related risk management. However, we believe our current setup effectively captures the primary trade-offs and dynamics needed to evaluate the policy. And we would like to start inviting the ML community to collaborate with us in further building and refining the simulator.
>
> Thank you again for your valuable feedback. Please advise whether our planned additional experiments address your concerns, and we welcome any further questions or comments you may have. Your insights are instrumental in helping us strengthen this work, and we look forward to your continued guidance during the rebuttal phase.
>
>
> Reference:
>
> [1] Miller, Zach. (2023, October 17) *Nearly Half of Fortune 500 Companies Engaged in Major Climate Initiatives. David Gardiner and Associates.* https://www.dgardiner.com/fortune-500-climate-initiatives-2023/
>
> [2] European Investment Bank, Kalantzis, F., & Cimini, F. (2023). What drives firms’ investment in climate action? : evidence from the 2022-2023 EIB investment survey, European Investment Bank.
>
> [3] Fortune global 500 climate commitments. Climate Impact Partners. (2024). https://www.climateimpact.com/news-insights/fortune-global-500-climate-commitments/
>
> [4] LSEG Data & Analytics. (2023) *Environmental, social and governance scores from LSEG.* https://www.lseg.com/content/dam/data-analytics/en_us/documents/methodology/lseg-esg-scores-methodology.pdf
>
> [5] Shek, Katia. (2023, January 30). *Patagonia: ESG's Golden Child. Global Research and Consulting Group Insights.* https://insights.grcglobalgroup.com/patagonia-esgs-golden-child/
>
> [6] InfluenceMap. (2022, September). *Big oil’s real agenda on Climate change 2022.* https://influencemap.org/report/Big-Oil-s-Agenda-on-Climate-Change-2022-19585
>
> [7] Buchner, Barbara. (2023, November 2). *Annual finance for climate action surpasses USD 1 trillion, but far from levels needed to avoid devastating future losses. Climate Policy Initiative.* https://www.climatepolicyinitiative.org/press-release/annual-finance-for-climate-action-surpasses-usd-1-trillion-but-far-from-levels-needed-to-avoid-devastating-future-losses/
>
> [8] Damodaran, Aswath. (2024, January). *Capital Expenditures by Sector.* https://pages.stern.nyu.edu/~adamodar/New_Home_Page/datafile/capex.html

---

> > ### Author Response · Authors · 2024-11-26
> >
> > Thank you again for your valuable feedback. As promised, we wanted to share the results of experiments we conducted with the following modifications: (1) an increased number of agents (25 companies and 25 investors), (2) company agents seeded with real-world company actions, and (3) investment commitments locked in for five years, and (4) adding random noise to determine the financial cost of extreme weather events which varies across companies and events. For company agents seeded with real-world company actions, we referred to the publicly available data from authority sources such as European Investment Bank and London Stock Exchange to seed the initial climate investment for 5 years, to resemble investment happening in the real-world. Across all these experiments, we observed the same directional results as in our original experiments. That is, in the default case where investors are only profit motivated, companies do not learn to mitigate effectively, climate risks remain high, and market wealth is lower. However, by including investors which place a high weight on ESG scores, companies can learn to mitigate, decrease climate risk, and increase total market wealth. Since the system does not permit us to share figures here, we will include these findings in the updated version of our paper, which will be uploaded by 11/27. Additionally, we really appreciate the relevant papers you have shared with us in your initial response, and we found some of them really informative and included them in our citation.
> >
> > The “adding random noise to determine the financial cost of extreme weather events” experiment was based on our actively engaging discussion with reviewer fPY9, which *may help address your concern around bankruptcy* as well. Our current framework already forces companies to go bankrupt if their capital becomes negative. However, in the experiments conducted so far, bankruptcy has not occurred because companies maintain an underlying economic growth rate sufficient to offset climate-induced losses and have learned not to overspend on climate-related efforts. In a new experiment, we will make companies’ financial losses during climate events highly stochastic, so that climate events could become severe enough to trigger bankruptcies. In addition, we made the standard for determining whether a company is bankrupt more realistic by deeming companies that have a margin worse than 10% for 3 consecutive years as bankrupt. This adjustment could reveal more insights on what happens if companies go bankrupt.
> >
> > As we approach the rebuttal deadline, we would greatly appreciate any additional feedback you can provide on how to further strengthen our paper.

---

> > ### Comment · Reviewer_mQaB · 2024-11-26
> >
> > Thank you for the detailed responses to my and other reviewers' comments and suggestions, including descriptions of additional proposed experiments, of the JAX implementation, and the plan to include a link to the JAX-based implementation in the final version of the paper.
> >
> > Based on these, the paper seems to have greatly improved, and I look forward to reading the new version, once submitted.
> >
> > As all the reviewers commented, the main strength of the paper is its potential contribution on a timely and crucial topic. The main aspects I will focus on when reviewing the new version of the paper include: (i) advances towards more realistic agent behaviour and scenarios; (ii) model validation; (iii) quantitative results; (iv) discussions and critical evaluation of the assumptions, results and InvestESG. Ultimately, does the paper provide sufficient evidence for the statement in the abstract: "However, when a critical mass of investors prioritizes ESG, corporate cooperation increases, which in turn reduces climate risks and enhances long-term financial stability. "?

---

> > > ### Author Response · Authors · 2024-11-28
> > >
> > > Thank you for engaging with us and sharing your specific concerns when reviewing the new version of the paper. As we promised, we have included a JAX-based implementation of the environment in the supplementary material.
> > >
> > >  >Advances towards more realistic agent behaviour and scenarios.
> > >
> > > In response to your valuable feedback, we have made a series of changes to enhance our model and capture more caveats and complexity in the real world in our rebuttal revision, which you can find more details in in Appendix 11.
> > > - Scaling up the number of agents: We scaled up the number of agents in the environment, conducting experiments with 10-company, 10-investor setups and larger 25-company, 25-investor setups. These scaled scenarios yield directionally consistent conclusions with those from smaller setups.
> > > - Real-World Initializations: We initialized company agents with real-world corporate actions to better align with actual decision-making contexts.
> > > - Capital Flexibility Challenges: We introduced a setting that locks in agents' decisions for five years, reflecting the inertia in capital allocation decisions faced by companies and investors in the real world.
> > > - Unpredictability of Climate Event Damage: We introduced a setting in which companies have a randomized climate resilience parameter that varies across events and agents, simulating the unpredictability of climate event damages.
> > > - Bankruptcy Mechanism: We implemented a stricter bankruptcy mechanism, creating a more severe risk for company agents, which in turn affects systemic outcomes.
> > >
> > > > model validation:
> > >
> > > The series of new experiments above serve as evidence for the robustness of the model under different scenarios. The results are consistent with theoretical expectations from the literature, reinforcing the model's validity as a benchmark for studying climate investment dynamics.
> > >
> > > > quantitative results;
> > >
> > > We have included additional social outcome metrics in the main text, including the average number of severe climate events in Figure 2 and Figure 4. We also include the number of companies that undergo bankruptcies in our additional experiments concerning bankruptcy mechanisms in Appendix 11.6.
> > >
> > > > discussions and critical evaluation of the assumptions, results and InvestESG.
> > >
> > > To clearly capture the assumptions we have made, we added a Problem Setting section, which includes the key trade-offs faced by the corporation concerning climate action, and how our model is grounded from well-established finance and economics literature that explores similar questions. We carefully presented and interpreted our results in the Main Result section as well as in Appendix 11, and compared them to theoretical and empirical literature, demonstrating compatibility with existing studies on ESG investment dynamics.
> > >
> > > We would like to clarify that, as added in the Introduction and Discussion section, we position InvestESG as a “first-principles” model that highlights the core incentive structures of the problem, rather than to fully capture real-world complexities and details. We plan to add more features. But our goal is to develop an environment that is flexible for researchers to select the level of granularity that best suits their needs.
> > >
> > > > Ultimately, does the paper provide sufficient evidence for the statement in the abstract: "However, when a critical mass of investors prioritizes ESG, corporate cooperation increases, which in turn reduces climate risks and enhances long-term financial stability. "?
> > >
> > > In our revision, we included the results from experiments with 10-company, 10-investor setups and larger 25-company, 25-investor setups in Figure 8. These new results all agree on the above statement. We also provide additional evidence in Appendix Figure 11c showing that, when a critical mass of investors prioritizes ESG, the cooperating companies attract the majority of ESG-conscious investment, and form a positive feedback loop where their better margins draw in more investments.

---

### Official Review · Reviewer_9btN · 2024-11-08

**Soundness:** 2
**Presentation:** 3
**Contribution:** 2
**Rating:** 5
**Confidence:** 4

**Summary:**

The paper concentrates on developing a multi-agent reinforcement learning framework (which they name InvestESG), to be used to student individual and collective outcomes from company investment and climate risk. It is an overall well written paper on a timely and relevant problem, for which we clearly need to better understand how to drive investors decisions to align individual and collective objectives. The paper is completely application-driven, in the sense that the authors do not develop new methodology, or a new solution approach. They mainly focus on describing how the framework should look like for the purpose of the application. They then generate a lot of simulation results to study various aspects of the problem (e.g., greenwashing, different levels of ESG consciousness, etc.)

**Strengths:**

The main strength of the paper is the topic is focuses on, and the idea to bring some momentum towards the development of a general platform to simulate a multi-agent system with focus on climate risk and company behaviour. Another strength (but which may also be seen as a weakness - see below) is that the framework is simple, in the sense that it is easily interpretable and flexible enough to interact with e.g. policy makers. The authors also aim to produce some relevant results, which may be seen as of value by policy-makers.

**Weaknesses:**

As mentioned by authors in a last part of the paper, maybe the main weakness is the simplicity of the framework, which may prevent a broad audience from accepting that it realistically model a real-world situation, and that it may be ring some relevant insights to be used as input to policy-making. In my opinion, it feels like an oversimplified and stylised approach where, depending on a few assumptions and a few modelling changes, we could get the model to do completely different things. Therefore, I believe that quite more work is necessary for such a paper, starting with the importance of the underlying assumptions, assessing the impact of modelling choices, sensitivity analyses, etc. I am not critical of the fact the authors are engaging in such developments - I am saying instead that I feel more work is necessary before sharing this work/paper with the world.

**Questions:**

I think some of the key points to consider are:
- questioning assumptions, e.g. rationality of the agents, why they would behave as if employing RL, etc.
- convincing us that simulating a system with a limited number of agents provides us with insights that are relevant for systems for very large number of agents
I clearly recognise that such issues may be more generally valid for the case of MARL environments and broader than for the case of this paper only. However, here, in view of the importance of the application, I find these issues particularly relevant.

---

> ### Author Response · Authors · 2024-11-16
> **Response to Reviewer 9btN**
>
> Thank you for your valuable feedback in helping us improve our work. In response to your comments, we have provided additional explanations below to address your concerns. We would greatly appreciate any further feedback on whether these revisions effectively resolve your concerns, as we are committed to strengthening the paper through the rebuttal process.
>
> **System with a limited number of agents**: Thank you for raising the question of whether a limited number of agents can represent a market with thousands of real agents. Post submission, we have developed a GPU-efficient Jax version of our environment, which runs at 10x the previous speed, enabling us to scale the simulation significantly. We will run larger-scale scenarios (on the order of 50 agents) to test whether we obtain similar findings when scaling to more agents. We will share updates during the discussion period as soon as the new results are available. We will also include a link to the Jax-based implementation in the final version of the paper.
>
> **Additional scenarios**: We plan to add a few additional scenarios to capture more of the real-world complexity, including seeding the first few steps of company actions with real-world company behaviors, and implementing a lock-in period for investments.
>
> **Simplicity of the framework**: We acknowledge, as you noted, that our work uses a simplified framework. Considering the magnitude and complexity of the global market, there might never be a model that fully captures the climate-market dynamics. But we believe that our choice of model captures the most important trade-offs. Briefly, our reasons for this approach are as follows.
> - **It is grounded in the economics and finance literature** that studies similar questions with theoretical models. For example, in a recent highly impactful study published in a top finance journal, Pastor et al. (2021) built an analytical model for a single-period equilibrium that examines firm-investor tradeoffs much similar to ours. In their model, firms choose the “greenness” level, which affects their cost of capital, while investors select portfolios to maximize utility derived from both financial returns and their “taste” for green assets. Other influential studies in financial economics use comparable or even simpler frameworks. For example, Pedersen et al. (2021) model a single-period equilibrium with investors differentiated by their types of ESG preference, assuming fixed ESG characteristics for firms. Going beyond these works, we provide a modeling tool that can study a system which evolves over a much longer time period (100 years), rather than a single timestep.
> - **Our simulations agree with the key findings in these papers** suggesting that ESG-conscious investors prioritize green companies and promote positive social impacts by shifting investment towards green firms.
> - Our framework, albeit still simple, **allows for investigating more realistic and complex settings than the existing financial economics literature**. These include (1) climate evolution, where companies mitigate emissions to attract investment and reduce long-term climate risk exposure—often omitted in financial studies (2) the potential for greenwashing, linked to information asymmetry (e.g., Lyon and Maxwell, 2011), and (3) a dynamic multi-agent game where firms and investors interact over many periods, which is more realistic. The equilibrium of such a setting is challenging to solve analytically or numerically (Pakes and McGuire, 2001), and our simulations provide insights here.
> - Despite the simplicity of the framework, **our simulation results match real-world data collected from countries that have implemented an ESG disclosure mandate** of some sort. For example, the mandate encourages more truthful mitigation than greenwashing (Fiechter et al., 2022), as shown in Figure 6. Raising public awareness motivates positive actions (Delmas and Toffel, 2008; Bowen, 2000), which we show as Figure 7, where including more information about climate risks helps both corporations and investors resolve the dilemma (we note that the legend lines for with vs. without climate information are accidentally mislabeled (flipped) on 7b and will fix it in the revised PDF). While we discuss these connections in Section 4, we will make this more clear in the revised version. We hope that by showing our benchmark matches existing empirical evidence, but enables studying novel policy algorithms, it can provide a useful tool for attracting ML researchers to develop algorithms that can help resolve the social dilemma posed by climate change investment.
> - For these reasons, we deliberately aimed to make a benchmark that is **simple enough for the ML community to iterate on with reasonable computational resources** (even a single GPU), without compromising the fundamental incentive structures. We believe removing the computation barrier will help us encourage greater participation from the ML community.

---

> > ### Comment · Reviewer_9btN · 2024-11-24
> >
> > Thanks for your reply, and your willingness to make changes to your paper.

---

> > > ### Author Response · Authors · 2024-11-26
> > >
> > > Thank you for following up. We wanted to share the results of additional experiments we conducted with the following modifications based on our engaging discussions with Reviewer fPY9: (1) an increased number of agents (25 companies and 25 investors), (2) company agents seeded with real-world company actions, (3) investment commitments locked in for five years, and (4) adding random noise to determine the financial cost of extreme weather events which varies across companies and events. For company agents seeded with real-world company actions, we referred to the publicly available data from authority sources such as European Investment Bank and London Stock Exchange to seed the initial climate investment for 5 years, to resemble investment happening in the real-world. Across all these experiments, we observed the same directional results as in our original experiments. That is, in the default case where investors are only profit motivated, companies do not learn to mitigate effective, climate risks remain high, and market wealth is lower. However, by including investors which place a high weight on ESG scores, companies can learn to mitigate, decrease climate risk, and increase total market wealth. Since the system does not permit us to share figures here, we will include these findings in the updated version of our paper, which will be uploaded by 11/27.
> > >
> > > We would also like to address your question on why RL agents can approximate rational actors in the real-world. Our rationale behind this is: fundamentally, RL works by learning to estimate future rewards over many experiences with the environment. Although humans and companies cannot directly observe the future, they make decisions based on experience, reasoning, and expectations about how the world evolves, often with greater sophistication than RL agents trained from scratch over repeated episodes. RL agents mimic this process by iteratively learning from simulated episodes, which can be seen as analogous to the iterative learning process humans undergo through trial, error, and observation of historical patterns. For example, stock prices in financial markets reflect the aggregate expectations of rational agents regarding future economic conditions. Similarly, RL agents do not "know" the future; rather, they estimate expected outcomes by averaging over many plausible scenarios and optimizing their policies based on this probabilistic understanding.
> > >
> > > As we approach the rebuttal deadline, we would greatly appreciate any additional feedback you can provide on how to further strengthen our paper, or any additional concerns we could potentially address.

---

> ### Author Response · Authors · 2024-11-16
> **Response to Reviewer 9btN**
>
> (cont.)
> - We appreciate your point that the framework may seem oversimplified to the broader audience. **We will dedicate a new section in the updated paper** to discuss why we think our simulations are solid to help address these concerns.
>
> Thank you again for your valuable feedback. We hope our responses address your concerns, and we welcome any further questions or comments you may have. Your insights are instrumental in helping us strengthen this work, and we look forward to your continued guidance during the rebuttal phase.
>
>
> Reference:
>
> [1] Pástor, Ľ., Stambaugh, R. F., & Taylor, L. A. (2021). Sustainable investing in equilibrium. Journal of financial economics, 142(2), 550-571.
>
> [2] Pedersen, L. H., Fitzgibbons, S., & Pomorski, L. (2021). Responsible investing: The ESG-efficient frontier. Journal of financial economics, 142(2), 572-597.
>
> [3] Lyon, T. P., & Maxwell, J. W. (2011). Greenwash: Corporate environmental disclosure under threat of audit. Journal of economics & management strategy, 20(1), 3-41.
>
> [4] Pakes, A., & McGuire, P. (2001). Stochastic algorithms, symmetric Markov perfect equilibrium, and the ‘curse’of dimensionality. Econometrica, 69(5), 1261-1281.
>
> [5] Fiechter, P., Hitz, J. M., & Lehmann, N. (2022). Real effects of a widespread CSR reporting mandate: Evidence from the European Union's CSR Directive. Journal of Accounting Research, 60(4), 1499-1549.
>
> [6] Delmas, M. A., & Toffel, M. W. (2008). Organizational responses to environmental demands: Opening the black box. Strategic management journal, 29(10), 1027-1055.
>
> [7] Bowen, F. E. (2000). Environmental visibility: a trigger of green organizational response?. Business strategy and the environment, 9(2), 92-107.

---

### Author Response · Authors · 2024-11-28
**Notes on paper update**

We would like to inform the area chairs and reviewers that we have uploaded an updated version of our paper and supplementary materials with the following changes. We sincerely thank the reviewers for your valuable feedback, comments, and suggestions, as well as the time and effort dedicated to helping us improve our work.
- Implementation code in both PyTorch and JAX in supplementary materials.
- Additional experiments (presented in Appendix 11) which yield directionally consistent conclusions as our main text.
  - Scaled up the number of agents to 10-company, 10-investor, and 25-company, 25-investor
  - Initialized company agents with real-world company actions
  - Setting agents' decisions as locked-in for five years to simulate capital flexibility challenges
  - Setting companies' climate resilience parameter as random and vary across events and companies to simulate unpredictability of climate event damage
  - Implementing a more strict bankruptcy mechanism
- We have added a problem setting section to introduce the key trade-offs, and further grounded our work within existing economic literature. In the introduction and conclusion, we clarified that our goal is to develop a first-principle model that highlights the core incentive structures of the problem, rather than to fully capture real-world complexities. For future work, we plan to incorporate additional variations, including reviewer suggestions that could not be fully developed during the rebuttal phase. This will enable fellow researchers to select the level of granularity that best suits their needs.
- We reorganized the results section to present a clearer story. Additionally, we incorporated writing and presentation improvements and included references suggested by the reviewers. Some parts of the mathematical details of the environment are moved to the appendix.

---

### Author Response · Authors · 2024-12-01
**Global follow-up comment**

As the discussion period is coming to a close, we would like to provide further details and explanation of the 5 new experiments we ran during the rebuttal period to address reviewers’ suggestions and comments. We ask that reviewers please let us know if they have further questions or comments before discussion closes tomorrow.

**Scaled up the number of agents:**

*Requested by reviewers 9btN, mQaB, fPY9*

Figure 8 in Section 11 shows the results of scaling up our simulation from the original 5 companies and 3 investors (5+3) to both 10+10 and 25+25 agents. As is evident in Figures 8(a-b), we obtained results consistent with our original findings: when investors have the ESG-consciousness parameter set to 10, overall climate risk is significantly reduced compared to the default case where both investors and companies are purely profit motivated (no ESG-consciousness).

**Initialized company agents with real-world company actions:**

*Requested by reviewer mQaB*

Figure 9 (d-f) in Section 11 shows the effect of seeding the initial 5 years of the simulation with parameters based on statistics about real-world companies. Specifically, we seeded 50% of companies to invest between 0.5% and 1% of their total capital into mitigation, based on publicly available data on companies’ current mitigation efforts, as described in Section 11.3. Figure 9 (d-f) compares the results of this experiment with our original results (Status quo w/ mandate), and we see no significant differences as a result of this experiment. We hypothesize this is because in our original results companies explore randomly in the first few timesteps, and thus successfully explore mitigation as a strategy. However, they eventually learn not to mitigate because since there are no ESG-conscious investors, ‘defecting’ by not mitigating is the optimal strategy, as supported by the Schelling diagram in Figure 3(a).


**Set agents' decisions as locked-in for five years to simulate capital flexibility challenges:**

*Requested by reviewer fPY9*

Figure 9(g-i) in Section 11 shows the results of an experiment in which the allocation of capital is less flexible, and agents’ investment decisions are locked in for a period of 5 years. The figure compares our original results (Status quo w/ mandate) to the results of the lock-in experiment. Interestingly, we see that initial mitigation amounts are higher in the locked-in case, leading to lower climate risks for earlier training episodes. We hypothesize this is because companies initially explore mitigation, but it takes some time to learn that it is better to defect. We see that by the end of the training period, both experiments converge to similar values for the final mitigation amount, climate risk, and market wealth.

**Increase uncertainty in the amount of economic damage incurred by extreme weather events:**

*Requested by reviewer fPY9*

The reviewer made a strong case for the high level of uncertainty surrounding the economic damages of climate change (Farmer et al., 2015). Therefore we conducted an additional experiment in which economic losses from extreme climate events were modeled as Gaussian random variables $L^{C_i}_t \sim \mathcal{N}(\mu, \sigma), \quad \mu = 0.07, \ \sigma = 0.1$ clipped within the range [0,1], varying across both events and companies (described in Section 11.5). The results are shown in Figure 10, which explores several facets of this idea. Figures 10(a-c) show that when uncertainty is higher, climate risk remains higher, and overall market wealth is lower than it is in the default status quo case. This suggests it might be harder for companies to learn to mitigate in the face of such uncertain risks. Figures 10(d-g) investigate how uncertainty affects greenwashing in the presence of ESG-conscious investors. We see that when economic damage is more uncertain, companies invest more in greenwashing to attract immediate investment from ESG-conscious investors, while retaining a similar level of mitigation efforts. Figures 10(h-k) pertain to the stricter bankruptcy mechanism, as described below.

---

> ### Author Response · Authors · 2024-12-01
> **Global follow-up comment (cont.)**
>
> (cont.)
>
> **Bankruptcy graphs and a stricter bankruptcy mechanism:**
>
> *Requested by reviewers mQaB, fPY9*
>
> Although our original results did include the ability for companies to go bankrupt, since both reviewers mQaB, fPY9 were interested in how more companies going bankrupt might affect the results, we implemented a stricter bankruptcy mechanism, where if a company agent has a margin worse than -10% for 3 consecutive years, it is deemed as bankrupt (see Section 11.6). Figures 10(h-k) show the results. We see that even with this stricter bankruptcy mechanism, the results are almost identical to the original status quo case, where essentially no companies go bankrupt, there is little mitigation spending and high climate risk. However, interestingly when we combine the strict bankruptcy mechanism with more uncertain economic damage, we see that the number of bankrupt companies is significantly higher. In turn, companies invest significantly more in mitigation, leading to lower climate risk. We hypothesize that in the scenario where economic damage is more uncertain, companies are more likely to go bankrupt, so we hypothesize this incentivizes them to spend more on mitigation to avoid bankruptcy (which has much lower utility). This in turn leads to lower climate risks. However, the market wealth is also lower in the case where companies more frequently go bankrupt due to climate damage.

---

### Meta-Review · Area_Chair_dGwn · 2024-12-21

**Metareview:**

The paper introduces InvestESG, a multi-agent reinforcement learning (MARL) benchmark designed to study climate investment decisions as a social dilemma. The benchmark models incorporate decision-making in the presence of ESG-conscious investors under climate risk uncertainty. Agents balance profit-driven objectives with long-term climate resilience through investments in mitigation, greenwashing, and adaptive strategies. The paper presents extensive simulation results using scaled agent populations, dynamic financial shocks, and diverse policy scenarios.

All reviewers acknowledged the importance and potential high impact of the research question. The simulation results are consistent with real-world data and could provide valuable insights for policymakers. On the other hand, given the significance of the domain, the main concerns raised by the reviewers are whether the simulated platforms generalize and scale. Regarding generalizability, the authors point out that their assumptions are grounded in economic and finance models and that the results align with real-world observations. However, traditional economic models often provide additional insights into their results (e.g., conditions under which the outcomes hold). This aligns with 9btn’s concern on the current work: “Depending on a few assumptions and a few modeling changes, we could get the model to do completely different things.” It might be helpful to include a discussion and results on the robustness of the findings—for example, how sensitive the results are to modeling choices. Regarding scalability, the authors have promised to provide additional results, e.g., with more agents.

Overall, this is a borderline paper that could go either way. Due to its potential high impact, I lean towards recommending acceptance. If accepted, I strongly suggest that the authors address the reviewers’ comments in their revision.

**Additional Comments On Reviewer Discussion:**

The main points are summarized above. One reviewer is clearly in support of the paper, while the other two are more reserved. They mentioned that they won't object the paper being accepted if there is space but are not enthusiastic.

I am tentatively recommending acceptance, though the paper could reasonably go either way.

---

### Decision · Program_Chairs · 2025-01-22

Accept (Poster)